# Extracellular matrix rigidity controls breast cancer metastasis via TYK2-mediated mechanotransduction

Zhimin Hu [1,7], Hannah E. Majeski[1,7], Aida Mestre-Farrera[1], Shirong Cai[2], Arya Lalezarzadeh [1], Yichi Zhang [1], Kei-Ichiro Arimoto[3], Dong-Er Zhang[3,4], Helen Piwnica-Worms [2], Laurent Fattet [1,6] & Jing Yang [1,5] ✉

Mechanical cues from the extracellular matrix (ECM) regulate various cellular processes. In breast cancer, increased tumor stiffness is associated with elevated metastasis risks and poor survival. Here we report a unique role of the JAK family kinase TYK2 in suppressing breast cancer metastasis under low ECM stiffness. Genetic or pharmacological inhibition of TYK2 in mammary acini and patient-derived organoids leads to invasion at low stiffness by promoting Epithelial-Mesenchymal Transition, which is independent of cytokine-induced JAK/STAT signaling. TYK2 blockade promotes metastasis in breast tumor cell- and patient-derived xenografts. TYK2 localizes at the plasma membrane via IFNAR1 association under low ECM stiffness, whereas high rigidity causes TYK2 cytoplasmic mislocalization and inactivation. Consistently, normal breast epithelium displays membrane-localized TYK2, whereas invasive breast tumors exhibit cytoplasmic TYK2. These findings uncover a TYK2-dependent mechanism by which ECM rigidity suppresses breast cancer metastasis and underscore the need for breast cancer screening in patients receiving TYK2 inhibitors.

Mechanical forces from the extracellular matrix (ECM) provide crucial cues to guide cell proliferation, migration, and differentiation during both normal development[1–5] and pathogenesis processes such as tumor invasion and metastasis[6–12]. During tumor progression, enhanced deposition and remodeling of tumor ECM could lead to a highly fibrotic tumor microenvironment with increased ECM stiffnesses than their surrounding normal tissues, as observed in various human cancers[13–15]. While some breast tumors exhibit similar stiffness as normal mammary tissues, higher ECM stiffness is observed in a subset of human breast tumors. The presence of stiffer tumors is associated with distant metastasis and poor survival in breast cancer

patients[7–9,16,17], highlighting the pivotal role of ECM mechanics in tumor invasion and metastasis.

TYK2 is a member of the JAK kinase family (JAK1, JAK2, JAK3, and TYK2). Like other JAK kinases, TYK2 constitutively binds to upstream cytokine receptors, such as IFNAR1 at the cell membrane and activates downstream STAT proteins upon cytokine-stimulated conformational changes of receptor complexes[18–21]. Although TYK2 is highly expressed in various normal epithelial cell types as shown in The Human Protein Atlas, its biological functions are almost exclusively studied in immune cells in the context of inflammation. Recently, TYK2 has gained attention as a therapeutic target in autoimmune diseases due to its

[1]Department of Pharmacology, Moores Cancer Center, University of California, San Diego, La Jolla, CA, USA. [2]Department of Experimental Radiation Oncology, University of Texas MD Anderson Cancer Center, Houston, TX, USA. [3]Department of Pathology, Moores Cancer Center, University of California, San Diego, La Jolla, CA, USA. [4]Department of Molecular Biology, University of California, San Diego, La Jolla, CA, USA. [5]Department of Pediatrics, University of California, San Diego, La Jolla, CA, USA. [6]Present address: Apoptosis, Cancer and Development laboratory, Centre de Recherche en Cancérologie de Lyon, INSERM U1052-CNRS UMR5286, Lyon, France. [7]These authors contributed equally: Zhimin Hu, Hannah E. Majeski. ✉e-mail: jingyang@ucsd.edu

critical role in inflammation in immune cells. Numerous completed or ongoing clinical trials are investigating TYK2 inhibitors for treating chronic autoimmune conditions like psoriasis and ulcerative colitis.[22]. However, the potential roles of TYK2 in epithelial tissues remain largely unexplored and could have significant implications for the clinical use of TYK2 inhibitors.

Human and mouse basal-like mammary epithelial cells, which are known to express a gene signature enriched with the Epithelial-Mesenchymal Transition (EMT) program, undergo EMT and invade surrounding matrices when cultured at tissue stiffnesses between 1000–5700 Pascals (Pa), which were detected in some human breast tumors[8–10,23]. In contrast, these same cells form mammary acini with adhenrens junctions and intact basement membrane under ~150–320 Pa observed in normal breast tissue even though they express endogenous EMT transcription factors[9,10]. Whether and how soft ECM stiffness suppresses breast cancer invasion and metastasis is not fully understood. Using human and mouse basal subtype mammary acini and patient-derived organoids in 3D and in orthotopic xenografts, we uncover a mechanotransduction pathway by which normal mammary tissue stiffness actively suppresses breast tumor invasion and metastasis via the TYK2 kinase.

## Results

### TYK2 is essential to suppress EMT and invasion at low ECM stiffnesses of normal mammary tissues

In a search for key molecular regulators linking ECM mechanical cues to EMT and invasion, we utilized a hydrogel Matrigel overlay 3D mammary acini culture system as the following: 1) the base layer is the polyacrylamide(PA) hydrogel with calibrated elastic moduli ranging from 150-320 Pa present in normal human breast tissues to 1100-5700 Pa observed in some stiff breast tumors; 2) the PA hydrogel is crosslinked with collagen I to enable mechanical coupling across the interface; 3) mammary acini embedded in Matrigel, which provides basement membrane-like matrices, are seeded on top of the hydrogel to be exposed to stiffness cues coming from the underlying collagen I-coated PA gel. The 3D model employed here recapitulates the biochemical and mechanical ECM microenvironment of breast carcinoma in situ, enabling mechanistic dissection of rigidity-dependent signaling events that initiate EMT. At low ECM rigidities, human basal-like non-tumorigenic MCF10A cells form mammary acini surrounded by basement membrane, while they undergo EMT and invade at high rigidities (Fig. 1a, f and Supplementary Fig. 1b). We treated these cultures with various kinase inhibitors, including inhibitors with differential specificities against members of the JAK kinase family: JAK1, JAK2, and TYK2 (JAK3 is not expressed in mammary epithelial cells)[18,24]. Surprisingly, treatment with momelotinib, an inhibitor targeting JAK1, JAK2 and TYK2[25], induced cell invasion at low matrix stiffnesses (320 Pa) (Fig. 1a, b); in contrast, treatment with filgotinib, which primarily targets JAK1 and JAK2[26], did not affect epithelial acini phenotype or cell invasion (Fig. 1a, c). Furthermore, treatment with either inhibitor had no effect on extensive cell invasion occurring at high stiffness of 5700 Pa (Fig. 1b, c). These data suggest a potential role of TYK2, but not other JAK family kinases, in suppressing EMT and invasion at low ECM stiffnesses.

To access the involvement of TYK2 in matrix stiffness-regulated EMT and invasion, we first knocked down endogenous human or mouse TYK2 in both human and mouse basal subtype mammary acini that are known to present EMT gene signatures and express endogenous EMT transcription factors[9,10,27]. In human MCF10A and Ras-transformed MCF10A derivative MCF10 DCIS acini cultures, knockdown of TYK2 potently induced EMT, as shown by increased Vimentin and decreased E-cadherin expressions[28] (Fig. 1f, g), and promoted invasion at low stiffnesses similar to normal mammary glands, while the control acini retained polarized features with membrane E-cadherin (Fig. 1d–g and Supplementary Fig. 1a). Laminin V (laminin-

332) immunostaining revealed a well-defined basement membrane structure surrounding acini under low stiffness, whereas this organized basement membrane layer was largely disrupted upon TYK2 knockdown (Supplementary Fig. 1b). It is worth noting that TYK2 knockdown did not drastically increase cell proliferation as detected by Ki67 staining (Supplementary Fig. 1c, d). To demonstrate the unique role of TYK2 in suppressing invasion is conserved across species, we also knocked down TYK2 in mouse EPH4Ras basal-like mammary tumor acini. Consistent with the human TYK2 results, knockdown of mouse TYK2 led to a significantly higher percentage of mammary acini invade at low stiffnesses compared to the control acini (Fig. 1h, i and Supplementary Fig. 1e). Together, our results strongly indicate that TYK2 plays an essential role in maintaining the epithelial state and suppressing EMT and invasion in both mouse and human mammary acini at low ECM stiffnesses.

We next tested whether knockdown of TYK2 promotes tumor invasion in patient-derived breast cancer organoids. We analyzed a collection of human triple-negative breast cancer (TNBC) patient-derived xenografts (PDXs)[29,30] based on the mRNA expression of the EMT transcription factor TWIST1 by RNA-sequencing and the expression and subcellular localization of TWIST1 protein in tumor cells by immunohistochemistry as our previous studies identified TWIST1 as an EMT transcription factor that mediates ECM stiffness-induced EMT and invasion[9,10,31]. PIM025 and PIM046 PDX tumors present TWIST1 in both the nucleus and cytoplasm of tumor cells (Supplementary Fig. 1f). We isolated primary tumor organoids from freshly harvested PDX tumors and cultured them in the 3D hydrogel Matrigel overlay system with varying stiffnesses from 150 Pa to 5700 Pa (Fig. 1j). Indeed, both PDX models exhibited increased invasion upon increasing matrix stiffnesses (Fig. 1l, m). Unlike cell line-derived acini that synchronously underwent EMT and invaded at 1100 Pa, a subpopulation of PDX-derived organoids remained non-invasive even at 1100 Pa and 5700 Pa. Such heterogenous responses to various ECM stiffnesses likely reflect the genetic and epigenetic heterogeneity of patient-derived tumor cell populations. TYK2 knockdown significantly promoted invasion in PDX-derived organoids at all stiffnesses (Fig. 1k–m). These results demonstrate an essential role of TYK2 in suppressing EMT and invasion in human and mouse breast acini and patient-derived breast cancer organoids.

### Pharmacological inhibition of TYK2 promotes EMT and invasion at low ECM stiffnesses

Several allosteric TYK2 inhibitors have recently advanced to clinical stages for autoimmune diseases. The distinct role of TYK2, but not other JAK family members, in mechanotransduction prompted us to test whether pharmacological inhibition of TYK2 kinase could induce EMT and invasion under low ECM stiffness. Supporting this hypothesis, tyrosine phosphorylation of both endogenous and overexpressed TYK2 decreased in acini cultured under increasing stiffnesses (Fig. 2a, b), indicating that TYK2 kinase activity decreases upon increases in ECM stiffnesses. Deucravacitinib is a highly selective allosteric TYK2 inhibitor approved for treating moderate to severe plaque psoriasis[32]. Both deucravacitinib and a JAK1/2 specific inhibitor AZD1480[33] effectively blocked IFNα-induced phosphorylation of STAT1 and STAT3 on Tyr705 (Supplementary Fig. 2a), demonstrating the effectiveness of both inhibitors[19,20,24]. Treatment of MCF10A and MCF10DCIS acini with deucravacitinib potently induced EMT and invasion at low stiffnesses, very similar to the phenotype caused by TYK2 knockdown (Fig. 2c, d and Supplementary Fig. 2c, d). In comparison, treatment with AZD1480 did not impact invasion in mammary acini under various stiffnesses (Fig. 2c, d). To further assess the effect of TYK2 inhibition on TNBC invasion, we tested a second TYK2 inhibitor, zasocitinib (Supplementary Fig. 2b), which is currently in a phase IIb clinical trial for psoriasis. Consistent with the effects observed with deucravacitinib, zasocitinib treatment also promoted

  

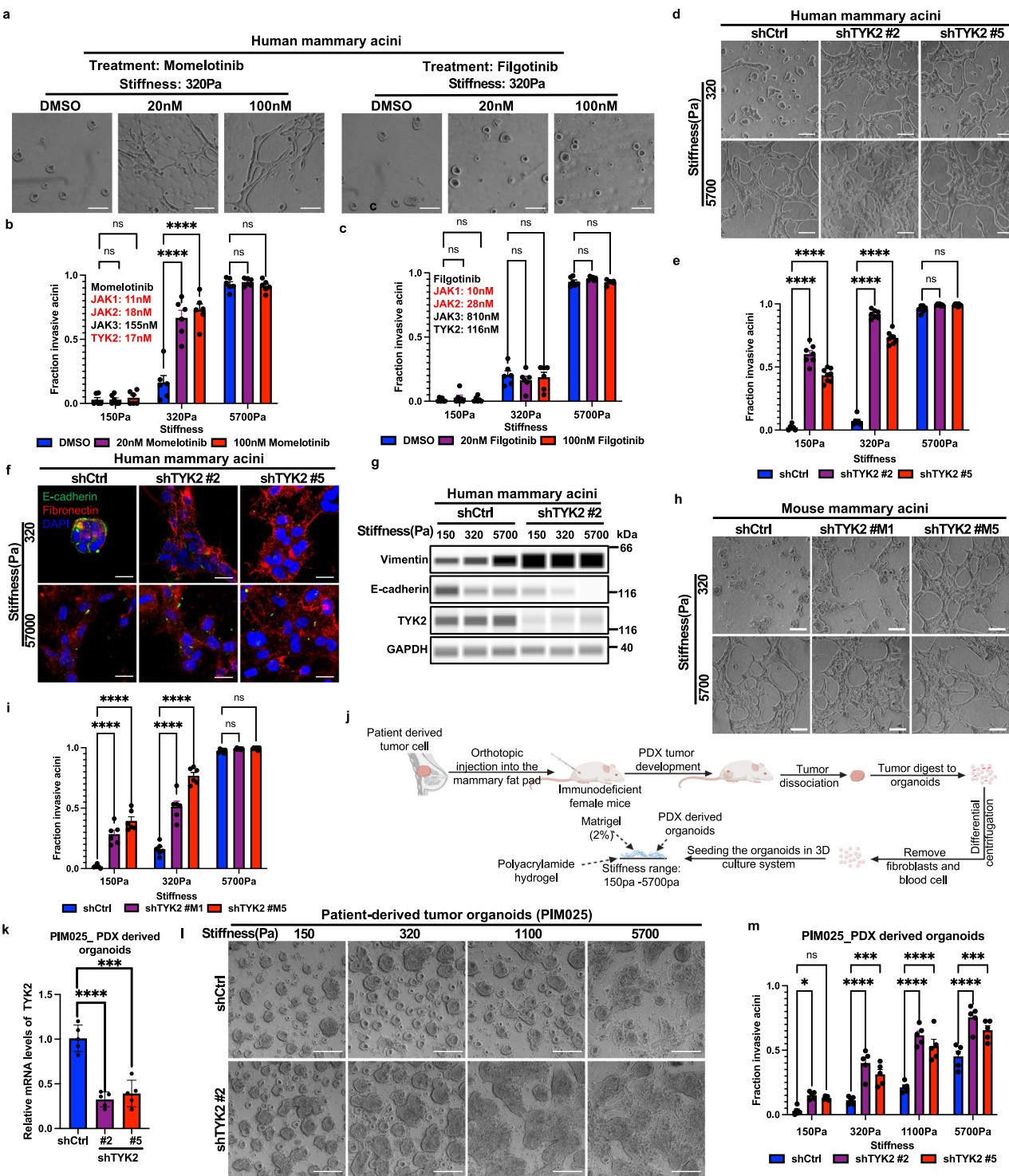

EMT and invasion under low stiffnesses (Fig. 2e, f). It is worth noting that TYK2 inhibition did not induce EMT due to compensatory activation of JAK1/2 as AZD1480 treatment still did not affect invasion in acini with TYK2 knockdown (Supplementary Fig. 2e, f). These results highlight a critical and unique role of the TYK2 kinase in suppressing EMT and invasion in human breast cancer cells.

We next examined whether pharmacological inhibition of TYK2 promotes tumor invasion in patient-derived TNBC organoids. Freshly isolated TNBC tumor organoids were treated with the TYK2 inhibitors deucravacitinib and zasocitinib in a 3D Matrigel overlay system. After five days of treatment, both inhibitors markedly increased

invasion, as tumor cells protruded into the surrounding matrices (Fig. 2g–j and Supplementary Fig. 2g, h). Again, treatment with a selective JAK1/2 inhibitor AZD1480 did not impact invasion (Supplementary Fig. 2g, h). Consistent with these effects, deucravacitinib decreased Tyr1054/1055 autophosphorylation of endogenous TYK2, altered the expression of EMT markers E-cadherin and vimentin, and further increased basal marker keratin 5(KRT5) at low ECM stiffnesses (Fig. 2k and Supplementary Fig. 2i). Together, these results demonstrate that pharmacological inhibition of TYK2 kinase activity promotes EMT and invasion under soft ECM stiffness in patient-derived TNBC organoids.

**Fig. 1 | TYK2 blockade in mammary acini promotes EMT and invasion under normal mammary tissue stiffness. a–c** Human MCF10A acini grown in three-dimensional polyacrylamide (3D-PA) hydrogels were treated for 5 days with momelotinib (JAK1/JAK2/TYK2 inhibitor), filgotinib (JAK1/2 inhibitor), or vehicle (DMSO) at the indicated concentrations. Representative bright-field images (**a**) and quantification of invasive structures (fraction of total) after momelotinib (20 nM, 100 nM) (**b**) or filgotinib (20 nM, 100 nM) (**c**) treatment (*n* = 6 wells per group, three independent experiments). **d–g** Control or TYK2-silenced MCF10A acini grown on 3D-PA gels for 5 days. Representative bright-field images (**d**), quantification of invasive structure (**e**) (*n* ≥ 6 wells per group, three independent experiments), immunofluorescence staining for E-cadherin (green), fibronectin (red), and DAPI (blue), representative images from 3 independent experiments (**f**), and immunoblot analysis of E-cadherin, vimentin, and TYK2 with GAPDH as loading control, representative of two independent experiments (**g**). **h, i** Control or TYK2-silenced mouse Eph4Ras acini grown on 3D-PA gels for 5 days. Representative bright-field images (**h**) and quantification of invasive structures (**i**) (*n* = 6 wells per group, three independent experiments). **j** Schematic of patient-derived organoid isolation and three-dimensional culture of PDX-derived organoids in PA hydrogels of defined stiffness (Created in BioRender. https://BioRender.com/dqo40p6). **k–m** Control or TYK2-silenced PDX-derived organoids (PIM025) grown on 3D-PA gels for 5 days: qPCR analysis of TYK2 mRNA expression, *n* = 5 wells per group, three independent experiments (**k**), representative bright-field images after 5 days on 3D-PA gels (**l**), and quantification of invasive structures (**m**) (*n* = 5 wells per group, three independent experiments). Invasive structures were scored using predefined morphological criteria (protrusive, stellate/branching outgrowth) and expressed as a fraction of total acini/organoids per sample. Data are mean ± SEM, dots represent independent wells. Where indicated, quantification is shown for different wells from one representative experiment, and similar results were obtained in independent replicate experiments (see n and number of independent experiments stated for each panel), ****$p$ < 0.0001; ***$p$ < 0.001; *$p$ < 0.05; ns, not significant. Two-group comparisons used unpaired two-tailed Student's t-test; multiple comparisons used one-way ANOVA with Dunnett's multiple-comparisons test. Scale bars: 100 μm (**a, d, h, l**), 25 μm (**f**). Exact *P* values and source data are provided in the Source Data file.

## TYK2 is required for maintaining TWIST1 cytoplasmic localization to prevent EMT and invasion at low ECM stiffness

We next sought to understand how TYK2 inhibits EMT and invasion under low matrix stiffness. High matrix stiffness is shown to promote TWIST1 nuclear translocation, thereby driving EMT-associated transcription programs[9]. We first examined the subcellular localization of endogenous TWIST1 protein in human MCF10A and MCF10DCIS, as well as mouse EPH4Ras mammary acini following TYK2 knockdown. At low stiffnesses, TWIST1 is predominantly localized in the cytoplasm; however, TYK2 knockdown induced aberrant nuclear localization of TWIST1, resembling the phenotype observed in acini cultured at high ECM stiffnesses (Fig. 3a, b and Supplementary Fig. 3a, b). Further supporting this result, pharmacological inhibition of TYK2 with deucravacitinib also promoted TWIST1 nuclear localization at low stiffnesses (Fig. 3c, d). Our previous studies demonstrated that TWIST1 is sequestered in the cytoplasm by its binding protein G3BP2 at low stiffnesses[9]. Therefore, we performed proximity ligation assay (PLA) to detect endogenous TWIST1/G3BP2 interaction in human mammary acini at low and high stiffnesses in response to TYK2 knockdown. In control acini, PLA signals were significantly enriched in the cytoplasm at low stiffnesses but was mostly lost at high stiffnesses. Knockdown of TYK2 resulted in loss of the PLA signal at low stiffnesses, indicating dissociation between TWIST1 and G3BP2 (Supplementary Fig. 3c, d). To further determine whether TYK2 is specifically involved in ECM stiffness-regulated TWIST1 translocation and EMT, instead of being required for maintaining epithelial properties downstream of EMT, we performed double knockdown of TWIST1 and TYK2 in MCF10A acini. Knockdown of TWIST1 completely abrogated EMT and invasion induced by TYK2 knockdown at 320 Pa (Fig. 3e–g and Supplementary Fig. 3e). Together, these results indicate that TYK2 functions upstream of TWIST1 to regulate its subcellular localization, thereby controlling EMT and invasion in response to ECM stiffness.

LYN kinase phosphorylates TWIST1 to disrupt its interaction with G3BP2 at high ECM rigidities[10]. Consistent with this mechanism, TYK2 knockdown significantly increased LYN kinase activity, indicated by Y416 phosphorylation on LYN (Fig. 3h). Pharmacological inhibition of LYN with bafetinib blocked TWIST1 nuclear localization and EMT induced by TYK2 knockdown at 320 Pa (Supplementary Fig. 3f, g). Furthermore, ligand-independent EPHA2 signaling via S897 phosphorylation is essential for matrix stiffness-induced LYN activation and TWIST1 nuclear localization at high stiffness[10]. TYK2 knockdown or TYK2 inhibition with deucravacitinib potently induced EPHA2 S897 phosphorylation at low stiffness (Fig. 3i–k). Mechanistically, we found that the active TYK2 kinase associated with EPHA2 at 320 Pa, but this interaction was markedly reduced at high stiffnesses of 1100 Pa and 5700 Pa, which coincided with increased S897 phosphorylation on EPHA2 (Fig. 3l). Taken together, these findings strongly suggest that TYK2 suppresses EMT and invasion by binding to EPHA2 and preventing S897 phosphorylation on EPHA2, thereby blocking LYN activation and TWIST1 nuclear localization at low ECM stiffness.

## High ECM stiffness leads to loss of TYK2 membrane localization, which is observed in invasive human breast tumors

To investigate how matrix stiffness regulates TYK2, we first examined TYK2 mRNA and protein levels under low and high matrix stiffnesses and found no significant changes (Supplementary Fig. 4a and Fig. 4a). Interestingly, in human MCF10A acini expressing FLAG-tagged human TYK2, TYK2 showed robust plasma membrane localization at low stiffness, but largely disappeared from the cell membrane at high stiffness (Fig. 4b, c). To demonstrate that this redistribution is due to stiffness sensing rather than cell morphological changes associated with EMT at high stiffness, we performed TYK2 immunostaining in single cells at low and high stiffnesses and found that TYK2 was localized at the cell membrane at low stiffnesses but became diffuse throughout the cell at high stiffness (Supplementary Fig. 4b, c). These data demonstrate that TYK2 subcellular localization is specifically regulated by ECM stiffness independent of epithelial polarity, cell-cell junctions, and epithelial morphology. Together, these data suggest that TYK2 localizes to the plasma membrane at low stiffness to suppress EMT and invasion but becomes diffusely distributed throughout the cytoplasm at high stiffness, thereby permitting EMT and invasion.

Given that TYK2 delocalization from the cell membrane is associated with EMT and invasion at high stiffnesses, we next examined TYK2 localization in normal human mammary tissues and breast tumors. In all six normal human breast samples, TYK2 exhibited strong membrane localization and colocalized with E-cadherin in ductal epithelial cells, closely resembling the pattern observed in low-stiffness 3D cultures (Fig. 4d, e and Supplementary Fig. 4d). We then analyzed a tissue microarray (TMA) of human invasive breast tumors across stages I, II, and III. Strikingly, in all 17 breast tumor samples, TYK2 signal was diffuse throughout the cell body and did not colocalize with membrane E-cadherin signal (Fig. 4d, e and Supplementary Fig. 4d). Independent immunohistochemistry images generated by the Human Protein Atlas[34,35] also exhibit diffuse TYK2 signal in all 21 invasive human breast cancer samples, while TYK2 presents pronounced cell membrane localization in normal human breast epithelium (Fig. 4f). Collectively, these results demonstrate that TYK2 localization changes from membranous in normal breast tissues to cytoplasmic in invasive cancerous tissues, suggesting inactivation of the TYK2 kinase in human breast cancer.

## β1-integrin engagement, but not FAK is required for high stiffness-dependent TYK2 inactivation and EMT induction

To investigate how matrix stiffness regulates TYK2 kinase activity and subcellular localization, we first examined the involvement of integrins

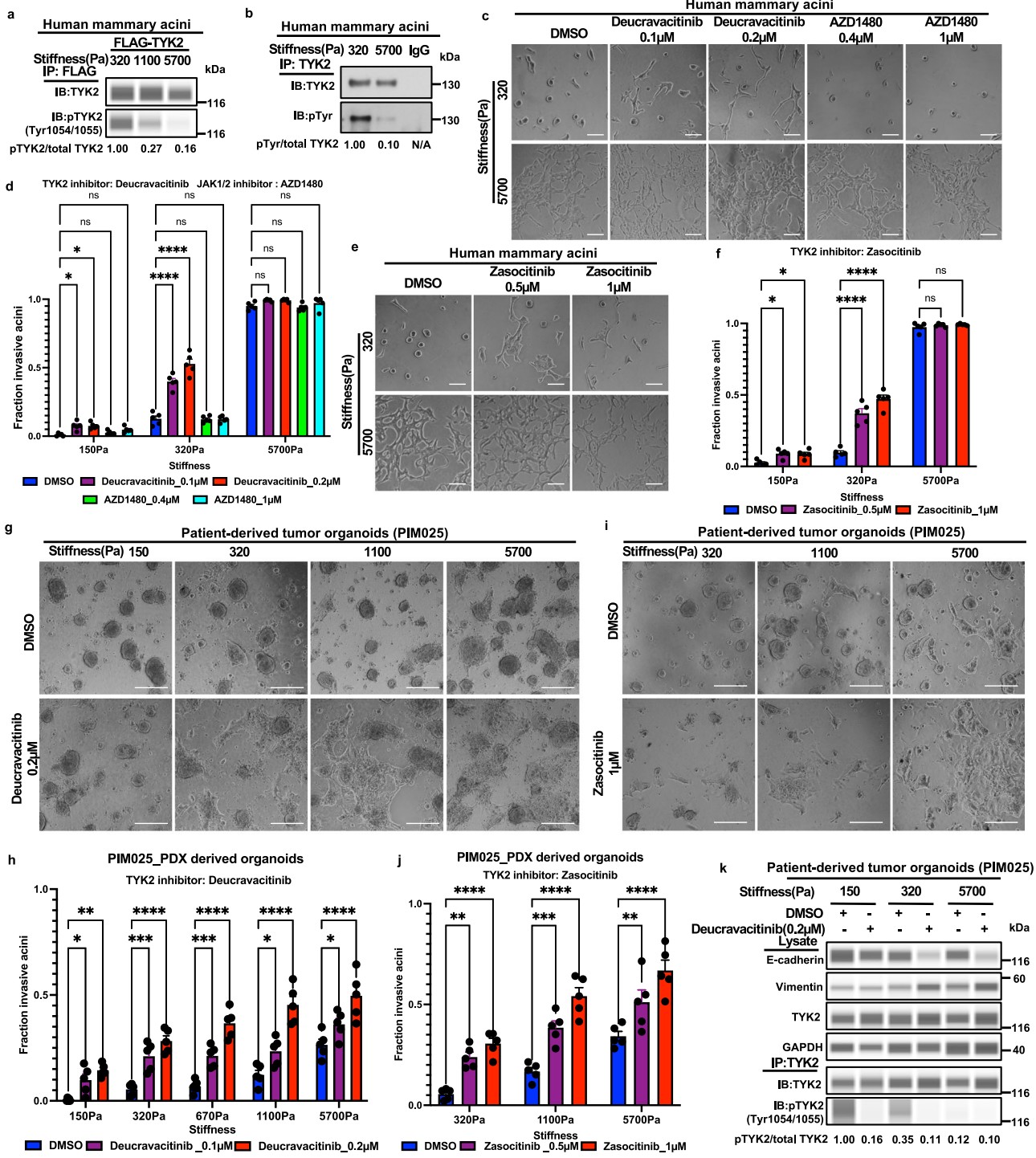

and FAK given their well-known roles in focal adhesion-mediated mechanotransduction. Treatment with a β1-integrin blocking antibody (AIIB2) that is known to block cell attachment to fibronectin, collagen-I, collagen-IV, and laminin[36] potently suppressed EMT and invasion by over 90% at all ECM rigidities (Supplementary Fig. 5a, b). Upon increasing ECM stiffness, cells undergo EMT and engage focal adhesion-mediated migration, as indicated by increased FAK phosphorylation at high rigidities (Supplementary Fig. 5c, e). However, treatment with the FAK inhibitor VS-4718 failed to block EMT at 5700 Pa, even though VS-4718 more potently suppressed FAK phosphorylation than the AIIB2 blocking antibody did (Supplementary Fig. 5c, d). Furthermore,

blockade of β1-integrin, but not FAK inhibition, prevented TYK2 inactivation indicated by phosphorylation of TYK2 on Tyr1054/1055 at high ECM stiffness (Supplementary Fig. 5e, f). Consistently, β1-integrin blockade also preserved TYK2 membrane localization at high ECM stiffness (Supplementary Fig. 5g, h). Together with what we reported previously that only β1-integrin blockade, but not FAK inhibition, could block TWIST1 nuclear localization and EMT at high ECM stiffness[9,10], these results suggest that integrin-β1 is essential for cells to bind to ECM and sense tissue rigidities. However, the FAK kinase is not responsible for TYK2 inactivation, but it functions to mediate mesenchymal cell migration upon EMT induction at high ECM stiffness.

**Fig. 2 | Pharmacological inhibition of TYK2 promotes EMT and invasion in human breast acini and patient-derived TNBC organoids. a, b** Lysates from 3D-PA gels cultured human MCF10A acini overexpressing FLAG-tagged wild-type TYK2 (**a**) or endogenous TYK2 (**b**) were subjected to anti-FLAG (**a**) or anti-TYK2 (**b**) immunoprecipitation, followed by immunoblot analysis of phosphorylated TYK2 (Tyr1054/1055) and total TYK2 (**a**) or phospho-tyrosine and TYK2 (**b**). IgG was used as a negative control, representative of two independent experiments. **c, d** Human MCF10A acini grown on 3D-PA gels were treated for 5 days with deucravacitinib, AZD1480, or vehicle (DMSO) at the indicated concentrations. Representative bright-field images (**c**) and quantification of invasive structures (fraction of total) (**d**) (n = 5 wells per group, three independent experiments). **e, f** Human MCF10A acini grown on 3D-PA gels were treated for 5 days with zasocitinib or vehicle (DMSO). Representative bright-field images (**e**) and quantification of invasive structures (**f**) (n = 5 wells per group, three independent experiments). **g, h** PDX-derived organoids (PIM025) grown on 3D-PA gels were treated for 5 days with deucravacitinib or vehicle (DMSO). Representative bright-field images (**g**) and quantification of invasive structures (**h**) (n = 5 wells per group, three independent

experiments). **i, j** PDX-derived organoids (PIM025) grown on 3D-PA gels were treated for 5 days with zasocitinib or vehicle (DMSO). Representative bright-field images (**i**) and quantification of invasive structures (**j**) (n = 5 wells per group, three independent experiments). **k** Lysates from PDX-derived organoids (PIM025) grown on 3D-PA gels and treated with deucravacitinib (200 nM) or vehicle for 5 days were subjected to anti-TYK2 immunoprecipitation followed by immunoblot analysis of phosphorylated TYK2 (Tyr1054/1055) and total TYK2; E-cadherin, vimentin, and TYK2 were analyzed in input lysates with GAPDH as a loading control, representative of two independent experiments. Data are mean ± SEM, where indicated, quantification is shown for different wells from one representative experiment, and similar results were obtained in independent replicate experiments (see n and number of independent experiments stated for each panel), ****p < 0.0001; ***p < 0.001; **p < 0.01; *p < 0.05; ns, not significant. Statistical significance was assessed using one-way ANOVA followed by Dunnett's multiple-comparisons test. Scale bars: 100 μm (**c**, **e**, **g**, **i**). Exact P values and source data are provided in the Source Data file.

## Matrix stiffness regulates TYK2/IFNAR1 interaction to control TYK2 membrane localization independent of the IFN/JAK/STAT signaling

We next focused on cytokine receptors, the known direct upstream regulators of TYK2. Given that inhibiting JAK1 and JAK2 did not affect stiffness-induced EMT and invasion, we first examined the expression of all known cytokine receptors that directly bind to TYK2[18,24,37] in human and mouse mammary cells and found that Interferon α/β receptor α chain (IFNAR1) is abundantly expressed (Supplementary Fig. 6a, b). Type I interferon signaling involves heterodimerization between IFNAR1 (a direct binder of TYK2) and IFNAR2 (a direct binder of JAK1) to activate downstream STAT transcription factors[19,21,38,39]. To test whether Type I interferon signaling is involved in TYK2 mechanotransduction, we first treated MCF10A acini with either human Type I IFN neutralizing antibody mixture or IFNα to block or activate the Type I interferon signaling and found neither treatment impacted acini morphology or invasion at various stiffnesses (Fig. 5a, b and Supplementary Fig. 6c, d). Consistent with this, treatment with a STAT3 inhibitor C188-9[40] and a STAT3/5 dual inhibitor SH-4-54[41] did not affect EMT and invasion under different ECM stiffnesses (Supplementary Fig. 6e–h). Together with the data in Figs. 1 and 2 demonstrating that inhibition of JAK1/2 does not impact stiffness-regulated EMT and invasion, these results suggest that the role of TYK2 in stiffness-regulated EMT and invasion is independent of the classical interferon/JAK/STAT signaling pathway.

While TYK2 suppresses EMT and invasion independent of its role in mediating JAK/STAT signaling, we next asked whether TYK2 needs to be anchored at the cell membrane by binding to IFNAR1 to perform its function in mechanotransduction. Strikingly, knockdown of IFNAR1, which directly binds to TYK2, resulted in similar invasion phenotypes as TYK2 knockdown at low ECM stiffnesses (Fig. 5c, e and Supplementary Fig. 6i–k); while knockdown of IFNAR2, which binds to JAK1, or knockdown of JAK1 did not alter cell invasion compared to the control cells at various stiffnesses (Fig. 5f,g and Supplementary Fig. 6l–p). Importantly, knocking down IFNAR1 in two human PDX organoid models PIM025 and PIM046 also led to invasion at low matrix stiffness (Fig. 5h, i and Supplementary Fig. 6q–s). Complement to these results, knockdown of IFNAR1 led to cytoplasmic redistribution of TYK2 protein, in contrast to its normal membrane association at low matrix stiffnesses (Fig. 5j, k). Knockdown of IFNAR1 also inhibited TYK2 kinase activity as indicated by decreases in Y1054/1055 phosphorylation and increased S897 phosphorylation on EPHA2 at 320 Pa (Fig. 5l). Co-immunoprecipitation revealed that TYK2 interacted with IFNAR1 and EPHA2 at 320 Pa, but this association was largely lost at higher stiffnesses (1100 Pa and 5700 Pa) (Fig. 5m). Moreover, knockdown of IFNAR1 resulted in loss of TYK2 binding to EPHA2 at 150 and 320 Pa (Fig. 5m). These results indicate that while TYK2 suppresses EMT and

invasion independently of canonical type I interferon signaling, TYK2 needs to be anchored at the cell membrane by IFNAR1 to perform its role in mechanoregulation of EMT and invasion.

## Genetic and pharmacologic inhibition of TYK2 promotes breast cancer invasion and metastasis

Since we found that TYK2 blockade induced EMT and breast tumor cell invasion at low stiffnesses, we next asked whether TYK2 functions as a metastasis suppressor to block breast cancer invasion and metastasis in vivo. We first utilized the human breast cancer MCF10DCIS GFP cells that recapitulate ductal carcinoma in situ with soft stiffnesses during the early stage of tumor development in the mammary fat pad. Once MCF10DCIS tumors progress into invasive tumors with larger sizes, force mapping by nanoscale atomic force microscopy (AFM) shows a > 10-fold increase in Young's modulus in some tumor areas compared to normal mammary glands[10,42]. We knocked down TYK2 in MCF10DCIS cells and injected them bilaterally into the mammary fat pad of immunodeficient female mice to allow tumor development for 6 weeks (Fig. 6a). While knockdown of TYK2 knocking did not affect primary tumor growth and weight (Fig. 6b and Supplementary Fig. 7a), it drastically increased the presence of GFP+ metastasis nodules in the lung (Fig. 6c, d). These data demonstrate that loss of TYK2 in breast tumor cells promotes breast cancer invasion and dissemination into the lung.

Recent clinical studies with the TYK2 inhibitor deucravacitinib reported high efficacy in treating psoriasis[43,44] and suggest the promise of TYK2 blockade in the treatment of various autoimmune diseases. However, our data suggest that TYK2 functions as a critical metastasis suppressor in TNBC in response to matrix stiffness. Therefore, we treated mice carrying MCF10DCIS early lesions with the TYK2 inhibitor deucravacitinib daily at a similar dose used in deucravacitinib preclinical studies in mice[45]. Consistent with TYK2 knockdown results above, both four and five weeks of treatment with deucravacitinib led to highly invasive primary breast tumors and significantly increased the number of lung metastases (Fig. 6e–j and Supplementary Fig. 7b–j). Deucravacitinib treatment reduced phospho-STAT3 signal in primary tumors, indicating effective inhibition of the TYK2 kinase activity (Supplementary Fig. 7b). Deucravacitinib treatment did not affect primary tumor growth or tumor weight (Fig. 6f and Supplementary Fig. 7c), which is also supported by similar Ki67 signal in both control and deucravacitinib-treated tumors (Supplementary Fig. 7d, e). Immunostaining showed a significant increase of nuclear TWIST1 signal in primary tumors treated with deucravacitinib in comparison to the vehicle-treated tumors (Fig. 6k, l). Consistent with the TYK2 knockdown results, knockdown of IFNAR1 in MCF10DCIS tumor cells significantly increased the number of lung metastases without affecting primary tumor growth (Fig. 6m–p and

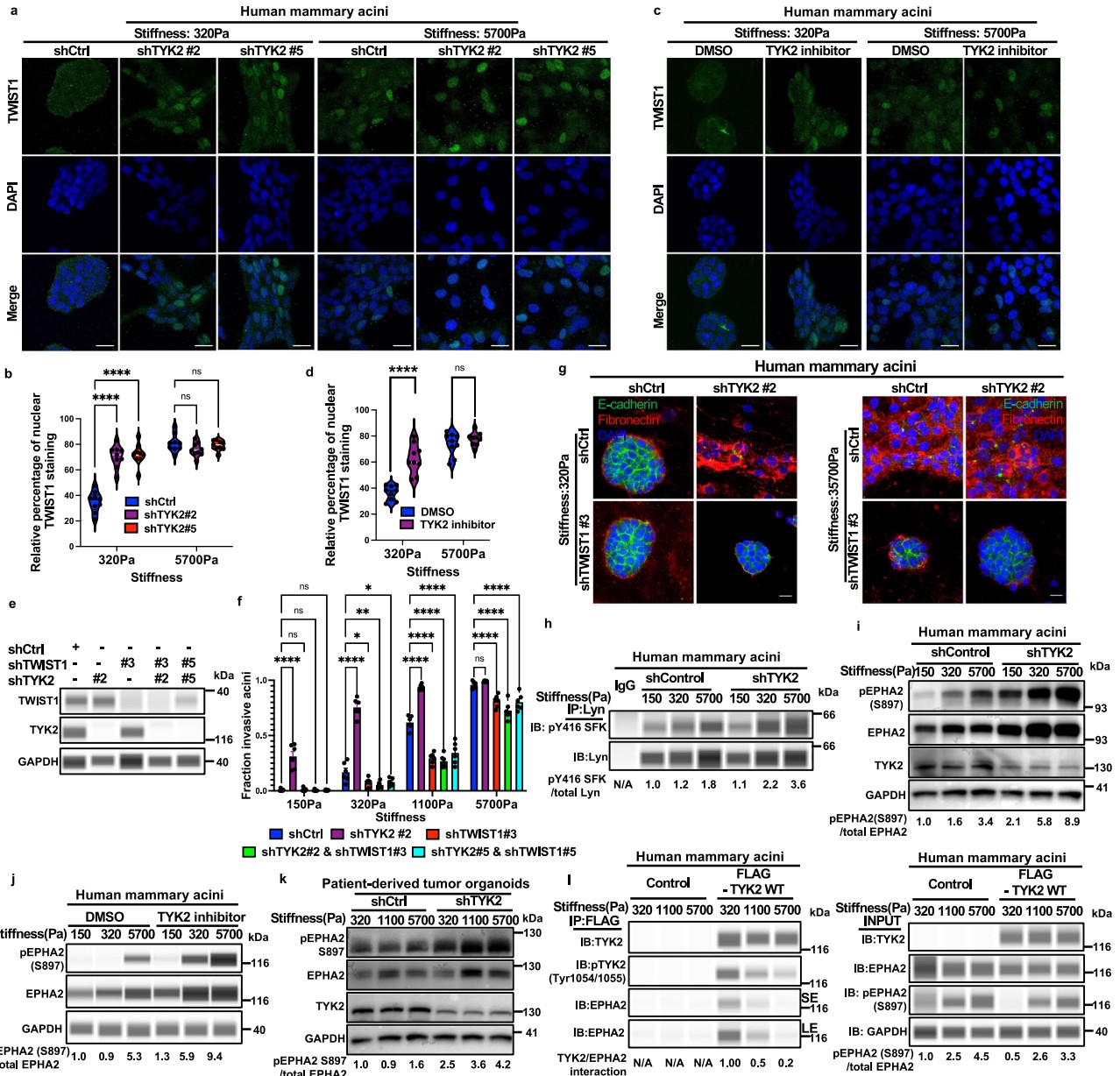

**Fig. 3 | TYK2 functions upstream of TWIST1 to control EMT and invasion.**
**a**, **b** Control or TYK2-silenced MCF10A acini grown on 3D-PA gels (5 days) were
immunostained for TWIST1 (green) and DAPI (blue) (**a**) and nuclear TWIST1 staining
was quantified as % of total TWIST1 staining (violin plots) (**b**). **c**, **d** MCF10A acini
grown on 3D-PA gels were treated for 5 days with TYK2 inhibitor deucravacitinib
(200 nM) or vehicle (DMSO) and immunostained for TWIST1 (green) and DAPI
(blue) (**c**); nuclear TWIST1 staining was quantified as % of total TWIST1 staining
(violin plots). Violin plots show the distribution of individual data points, the center
line indicates the median, the thick bar indicates the interquartile range (25th–75th
percentiles), each dot represents one measurement (n = 9 per condition) (**d**).
**e** Immunoblot analysis of TWIST1 and TYK2 in MCF10A cells with the indicated
shRNAs; GAPDH served as a loading control. Representative of two independent
experiments. **f**, **g** Control, TYK2-silenced, TWIST1-silenced, and TYK2/TWIST1
double-silenced MCF10A acini grown on 3D-PA gels for 5 days. Invasive structures
were quantified (**f**) (n= 5 wells per group, three independent experiments).
Representative immunofluorescence images are shown (**g**), stained for E-cadherin
(green), fibronectin (red), and DAPI (blue). Images are representative of two inde-
pendent experiments.**h** Lysates from control or TYK2-silenced MCF10A acini grown

on 3D-PA gels (5 days) were subjected to LYN immunoprecipitation followed by
immunoblot analysis,representative of two independent experiments.
**i–k** Immunoblot analysis of phosphorylated EPHA2 (Ser897), total EPHA2, and
TYK2 in lysates from control or TYK2-silenced MCF10A acini (**i**), MCF10A acini
treated with deucravacitinib (200 nM) or DMSO (**j**), or control or TYK2-silenced
PIM025 PDX-derived organoids (**k**) grown on 3D-PA gels (5 days), GAPDH was used
as a loading control, representative of two independent experiments. **l** MCF10A
acini expressing pWZL-Blast control or FLAG-tagged TYK2 were grown on 3D-PA
gels (5 days), and lysates were subjected to anti-FLAG immunoprecipitation fol-
lowed by immunoblot analysis, representative of two independent experiments,
SE short exposure and LE long exposure. Data are mean ± SEM, where indicated,
quantification is shown for different wells from one representative experiment, and
similar results were obtained in independent replicate experiments (see n and
number of independent experiments stated for each panel), ****p < 0.0001;
**p < 0.01; *p < 0.05; ns, not significant. Two-group comparisons used unpaired
two-tailed Student's t-test; multiple comparisons used one-way ANOVA with Dun-
nett's multiple-comparisons test. Scale bars: 25 μm (**a**, **c**, **f**). Exact P values and
source data are provided in the Source Data file.

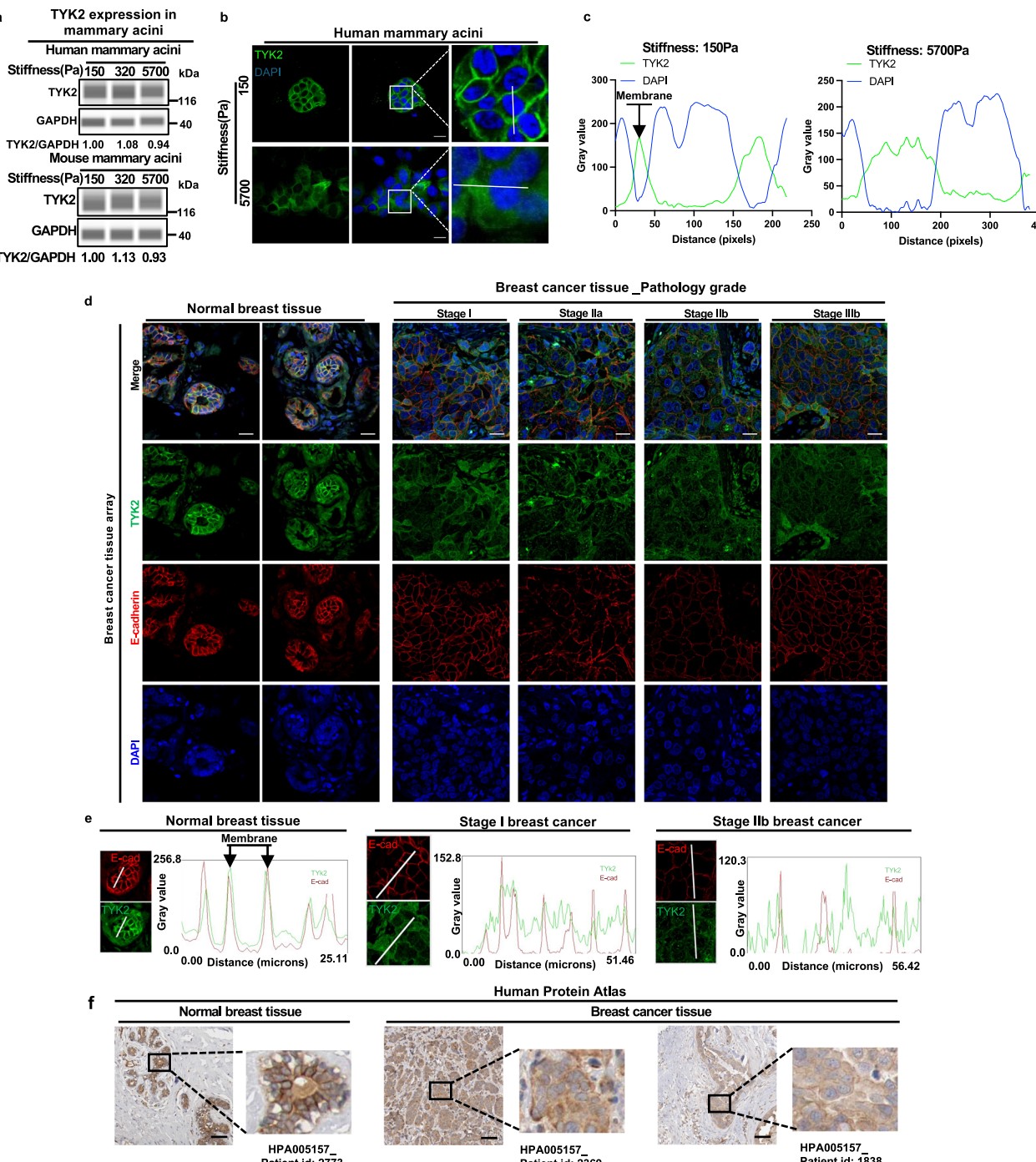

**Fig. 4 | Loss of TYK2 membrane localization at high matrix stiffness is associated with invasive human breast cancer. a** Immunoblot analysis of TYK2 protein expression in lysates from human MCF10A acini and mouse Eph4Ras acini grown on three-dimensional polyacrylamide (3D-PA) gels for 5 days, GAPDH was used as a loading control, representative of two independent experiments. **b** Immunostaining of TYK2 (green) and DAPI (blue) in human MCF10A acini overexpressing FLAG-tagged TYK2 or control cells grown on 3D-PA gels, representative images from 2 independent experiments. Scale bar, 25 μm. **c** Line scan analysis showing DAPI and TYK2 intensity profiles plotted as a function of distance across acinar structures in MCF10A acini overexpressing FLAG-tagged TYK2 or control cells grown on 3D-PA gels. **d** Immunostaining analysis of a human breast tissue microarray (BR248a) containing normal mammary tissues and invasive breast tumors of different pathological stages(6 cases of normal breast tissues, 17 cases of Invasive carcinoma, and 1 case of medullary carcinoma), stained for TYK2 (green), E-cadherin (red), and DAPI (blue), as indicated. Scale bar, 25 μm. **e** Line scan analysis of E-cadherin and TYK2 intensity profiles plotted as a function of distance across tissue sections from normal breast tissue and stage I and stage IIb invasive breast tumors in the tissue microarray. **f** Representative immunohistochemistry images of TYK2 in normal human mammary tissues and breast tumors, Scale bar, 100 μm. Images obtained from the Human Protein Atlas.(https://www.proteinatlas.org/ENSG00000105397-TYK2/tissue/breast). Source data are provided in the Source Data file.

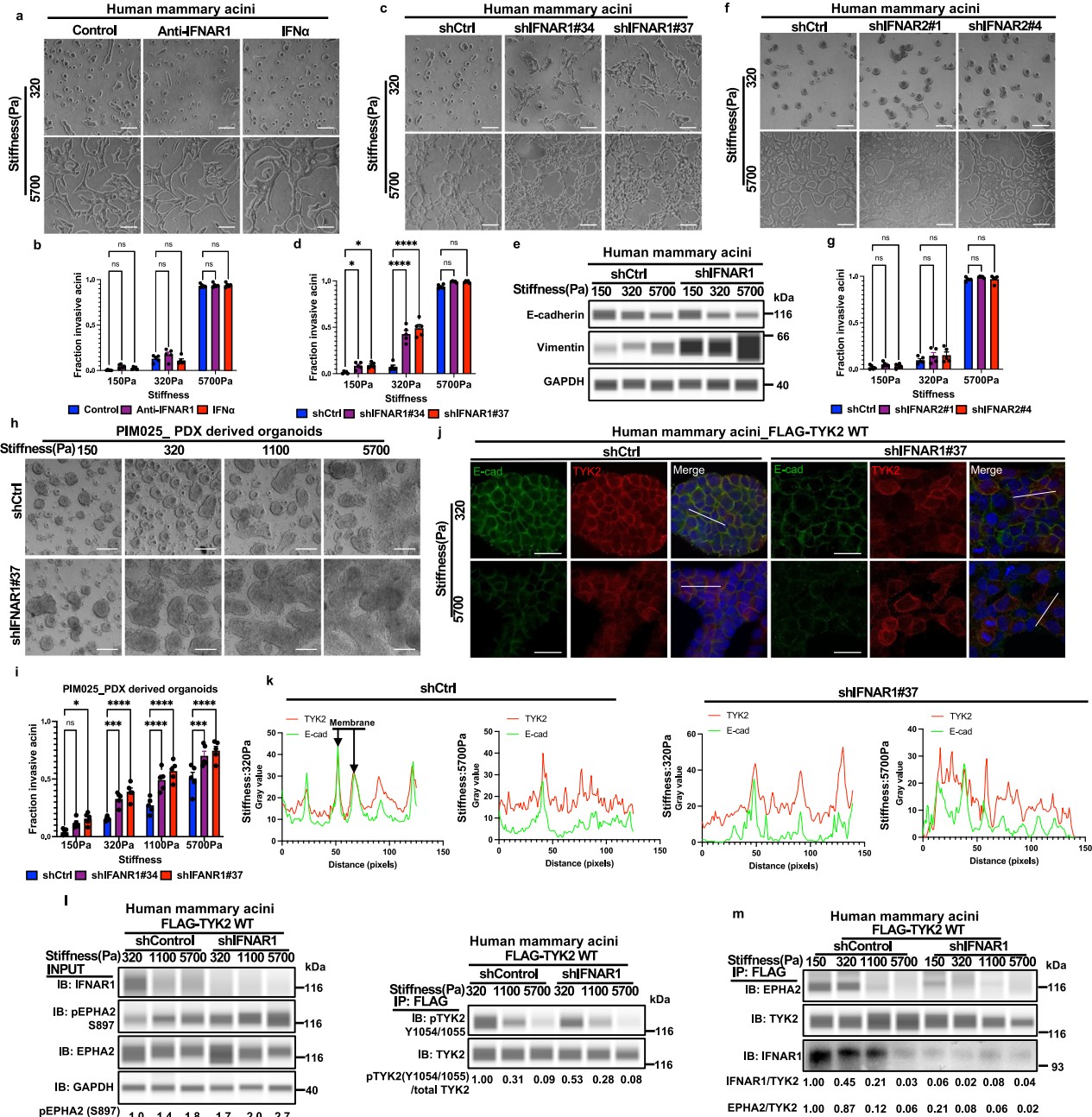

**Fig. 5 | IFNAR1 is essential to maintain TYK2 membrane localization to control EMT and invasion at low ECM stiffness. a, b** Human MCF10A acini grown on 3D-PA gels for 5 days were left untreated (control) or treated with an IFNAR1-blocking antibody or IFNα. Representative bright-field images (**a**) and quantification of invasive structures (b) ($n = 5$ wells per group, three independent experiments). **c, d** Control or IFNAR1-silenced human MCF10A acini were grown on 3D-PA gels for 5 days. Representative bright-field images (c) and quantification of invasive structures (**d**) ($n = 5$ wells per group, three independent experiments). **e** Immunoblot analysis of E-cadherin and vimentin in lysates from Control or IFNAR1-silenced human MCF10A acini grown on 3D-PA gels; GAPDH was used as a loading control (representative of two independent experiments). **f, g** Control or IFNAR2-silenced human MCF10A acini were grown on 3D-PA gels for 5 days. Representative bright-field images (**f**) and quantification of invasive structures(g) ($n = 5$ wells per group, three independent experiments).**h, i** Control or IFNAR1-silenced PIM025 PDX-derived organoids were grown on 3D-PA gels for 5 days. Representative bright-field images (**h**) and quantification of invasive structures (**i**) ($n = 5$ wells per group, three independent experiments). **j, k** Human MCF10A acini overexpressing TYK2

together with control or IFNAR1-targeting shRNAs were grown on 3D-PA gels for 5 days and immunostained for TYK2 (red), E-cadherin (green), and DAPI (blue), representative $n = 9$ images from 3 independent experiments. (**j**); line-scan analysis of E-cadherin and TYK2 intensity profiles plotted as a function of distance across acinar structures (**k**). **l, m** Human MCF10A acini overexpressing FLAG-tagged wild-type TYK2 together with IFNAR1-targeting or control shRNAs were grown on 3D-PA gels for 5 days, and lysates were subjected to anti-FLAG immunoprecipitation followed by immunoblot analysis as indicated (**l, m**) (representative of two independent experiments).Data are mean ± SEM, where indicated, quantification is shown for different wells from one representative experiment, and similar results were obtained in independent replicate experiments (see n and number of independent experiments stated for each panel), ****$p < 0.0001$; ***$p < 0.001$; **$p < 0.01$; *$p < 0.05$; ns, not significant. Two-group comparisons used unpaired two-tailed Student's t-test; multiple comparisons used one-way ANOVA with Dunnett's multiple-comparisons test. Scale bars: 100 μm (**a, c, f, h**), 25 μm (**j**). Exact *P* values and source data are provided in the Source Data file.

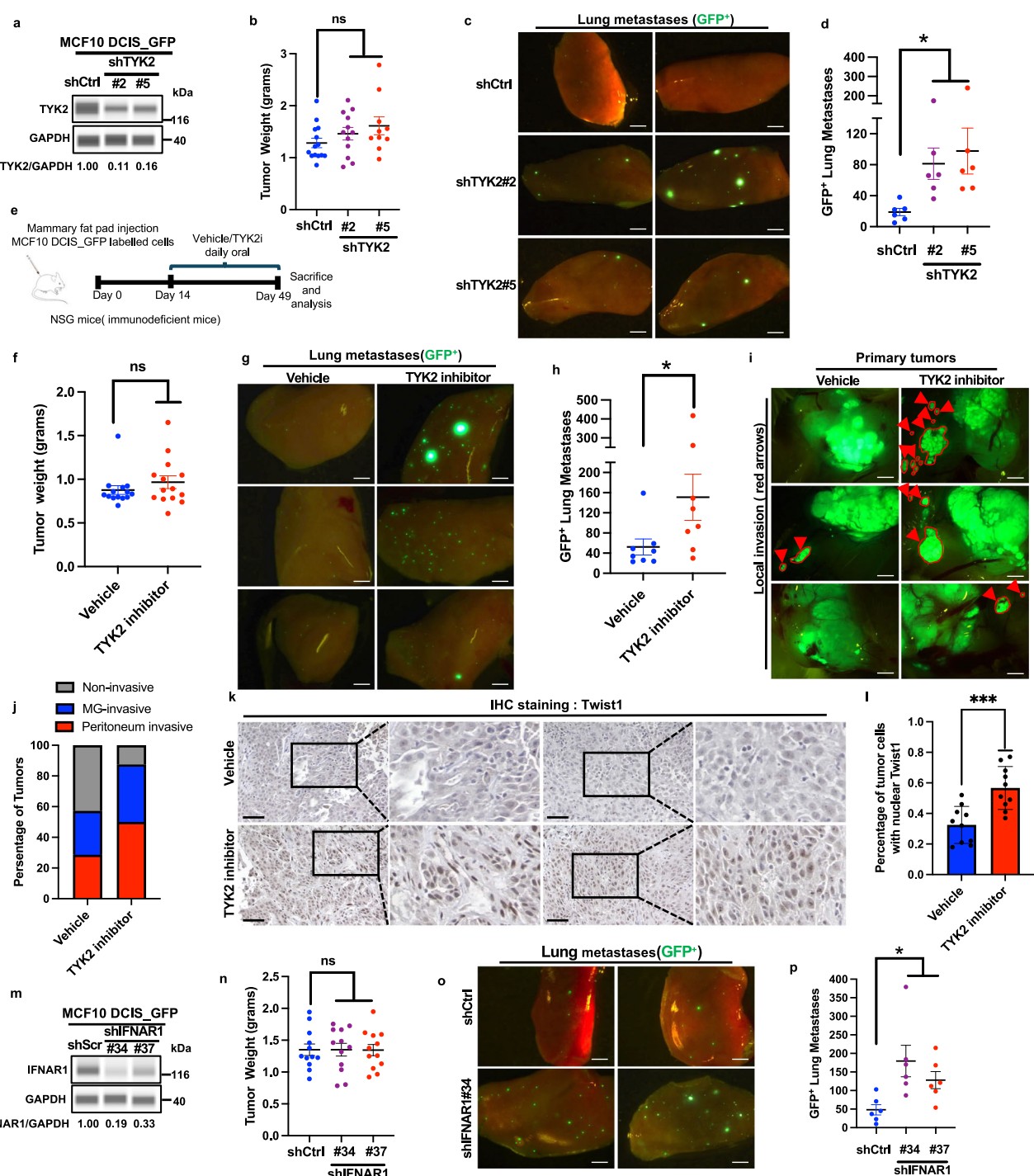

**Fig. 6 | TYK2/IFNAR1 blockade promotes breast cancer invasion and metastasis in vivo. a–d** Immunoblot analysis of TYK2 expression in MCF10DCIS primary tumors expressing control or TYK2-targeting shRNAs; GAPDH was used as a loading control, representative of two independent experiments (**a**). Tumor weight at the experimental endpoint, *n* = 12 tumor from 6 mice per group (**b**). Representative images of lung metastases from GFP-labeled MCF10DCIS xenografts, with GFP-positive metastatic cells shown in green, scale bar, 1.5 mm (**c**). Lung metastatic burden quantified as the number of GFP-positive nodules (**d**)(*n* = 6 mice per group). **e–j** Schematic of the TYK2 inhibitor treatment protocol: one million GFP-labeled MCF10DCIS cells were orthotopically injected into the mammary fat pad of female NSG mice, followed by daily oral gavage with vehicle or deucravacitinib for five weeks (**e**). Tumor weight at the experimental endpoint, *n* = 14 tumor from 7 mice per group (**f**). Representative images of lung metastases, scale bar, 1 mm (**g**) and quantification of GFP-positive lung nodules (**h**) (*n* = 7 mice per group).

Representative images of primary tumor invasion, with arrows indicating invasive tumor cells, scale bar, 2 mm (**i**), and percentage of primary tumors exhibiting local (mammary gland) or regional (peritoneum) invasion (**j**) (*n* = 7 mice per group). **k**, **l** Immunohistochemical analysis of TWIST1 localization in primary tumors, scale bar, 100 μm (**k**) and quantification of nuclear TWIST1 as the percentage of total cells (**l**) (*n* = 5 mice per group). **m–p** Immunoblot analysis of IFNAR1 expression in primary tumors expressing control or IFNAR1-targeting shRNAs, GAPDH was used as a loading control, representative of two independent experiments (**m**). Tumor weight at the experimental endpoint, *n* = 12 tumor from 6 mice per group (**n**). Representative images of lung metastases, scale bar, 1 mm (**o**) and quantification of GFP-positive lung nodules (**p**) (*n* = 6 mice per group). Data are mean ± SEM, ***p < 0.001; *p < 0.05; ns, not significant. Statistical significance was assessed using unpaired two-tailed Student's t-test with Welch's correction. Exact P values and source data are provided in the Source Data file.

Supplementary Fig. 7k, l). These data strongly support a tumor cell-autonomous metastasis suppressor function of TYK2 in basal-subtype breast cancer.

We next tested whether inhibiting TYK2 using deucravacitinib promotes tumor invasion and metastasis in patient-derived TNBC xenografts. We labeled freshly harvested PIM025 PDX tumor organoids with GFP and implanted them in the mammary fat pad of immunodeficient female mice (Fig. 7a–c) for 2–3 weeks followed by 4–5 weeks of TYK2 inhibitor treatment. TYK2 kinase inhibition did not affect primary tumor growth, tumor weight, and tumor cell proliferation marked by Ki67 signal (Supplementary Fig. 8a, c–h), but significantly increased the number of GFP+ lung metastases (Fig. 7d–i). The specificity of the TYK2 kinase inhibitor was demonstrated by diminished Y705 phosphorylation on Stat3 in primary tumors from deucravacitinib-treated mice (Supplementary Fig. 8b). Importantly, immunostaining of TWIST1 showed significantly increased nuclear TWIST1 signal in primary tumors treated with deucravacitinib (Fig. 7j, k). Furthermore, knockdown of TYK2 in PIM025 PDX tumor cells also promoted metastatic dissemination into the lung without affecting primary tumor growth (Fig. 7l–o). These results reveal that TYK2 is a key metastasis suppressor in human TNBCs and inhibition of TYK2 kinase promotes breast tumor invasion and metastasis.

## Discussion

Here we uncovered a potent metastasis suppression mechanism by which soft ECM stiffness suppresses EMT via the TYK2 kinase. Using human and mouse basal-subtype mammary acini and TNBC PDX-derived organoids, we show that this conserved function of TYK2 as a metastasis suppressor is unique to TYK2 and not shared by other JAK family kinases. Mechanistically, TYK2 functions to suppress TWIST1 nuclear translocation to inhibit EMT and invasion at low matrix stiffness in normal breast tissues. Importantly, we found that TYK2 suppresses EMT and invasion independent of the classical cytokine-stimulated JAK/STAT signaling. Upon interferon exposure, TYK2 normally heterodimerizes with JAK1 or JAK2 downstream of interferon receptors to phosphorylate and activate STAT proteins to induce interferon-responsive genes[18,24]. Treatment with Type 1 interferons (IFN-I) have been shown to improve distant-metastasis-free survival in patients with high-risk melanoma[46], suggesting a role in metastasis suppression largely by stimulating the host immune system[47]. Surprisingly, blocking IFN-I binding to its receptor, JAK1/2 kinases or their downstream STAT-dependent gene transcription had no impact on the role of TYK2 in mechanotransduction.

Instead, TYK2 plays an unexpected role in transmitting the mechanical cues from soft breast tissue ECM to suppress EMT and invasion. We show that at low matrix stiffness of normal mammary glands, TYK2 binds to IFNAR1 on the cell membrane and associates with EPHA2, thus blocking EPHA2 S897 phosphorylation. At high stiffnesses of breast tumor tissues, TYK2 dissociates from the cell membrane, thus allowing EPHA2 S897 phosphorylation and downstream activation of the TWIST1-driven EMT and invasion. These results suggest the mechanical cues from ECM lead to alternations of the IFNRA1/TYK2/EPHA2 complex on the cell membrane to generate downstream EMT transcription responses. Biochemical and biophysical studies are needed to uncover how mechanical forces exerted from the tumor ECM controls the IFNRA1/TYK2/EPHA2 complex, possibly via post-translational regulation of these proteins.

We found that while β1-integrin engagement is essential to control the TYK2 kinase activity in response to different ECM rigidities, the integrin downstream kinase FAK is dispensable for TYK2 mechanotransduction. This result is consistent with what we reported previously that β1-integrin, but not FAK is required for high stiffness-induced TWIST1 nuclear localization and EMT[9,10]. The YAP/TAZ transcription activators are also known to be regulated by ECM rigidities[48].

Interestingly, not only β1-integrin is required for YAP activation[49], but the FAK kinase has been shown to activate YAP nuclear localization via phosphorylation of MOB1 to inhibit core upstream Hippo signaling[50]. These distinctions demonstrate that FAK inhibition, which inhibits YAP/TAZ, does not impact TYK2/TWIST1 mechanotransduction and underscore that ECM rigidities impinge on YAP vs TWIST1 via distinct signaling regulators. Future work is needed to reveal how β1-integrin directly or indirectly impinges on TYK2/TWIST1 mechanotransduction in a FAK-independent manner in response to ECM rigidities.

Targeting TYK2 in immune cells has gained attention in auto-immune disease therapeutics. By selectively targeting TYK2, therapies may reduce inflammation and prevent immune cells from attacking healthy tissues, offering a potential treatment strategy for conditions like psoriasis, lupus, and other autoimmune disorders. Recent successful clinical trials with the TYK2 inhibitor deucravacitinib demonstrated high efficacy in treating moderate-to-severe plaque psoriasis, which affects 125 million people worldwide[43,44]. However, normal epithelial cells in many organs show abundant TYK2 protein on the cell membrane. Here we uncovered a potent metastasis suppression mechanism by soft ECM stiffnesses via the TYK2 kinase in TNBC cells. Treatment with two TYK2 inhibitors deucravacitinib or zasocitinib for five days led to EMT and invasive behavior in non-tumorigenic basal-subtype MCF10A acini and in patient-derived TNBC tumor organoids, and treatment of PDX xenografts for four weeks in mice drastically increased metastasis dissemination in the lung. Genetic deletion of TYK2 in TNBC tumor cells phenocopies the effects caused by TYK2 inhibitors, therefore the effect of TYK2 inhibition on tumor invasion and metastasis is on TNBC cells and independent of TYK2 functions in immune cells. These data not only demonstrate a critical role of TYK2 as a metastasis suppressor in TNBCs but also caution that patients with undiagnosed basal-subtype carcinoma in situ could be at risk of developing invasive breast cancer receiving TYK2 inhibitors. The metastasis suppression role of TYK2 in breast cancer uncovered here, together with the well-known role of TYK2 in immune cell activation, warrants future studies to determine the need of enhanced breast cancer screens in patients treated with TYK2 inhibitors.

## Methods
### Cell culture
MCF10A cells were grown in DMEM/F12 media supplemented with 5% horse serum, 20 ng/ml human EGF (hEGF), 10 mg/ml insulin, 0.5 mg/ml hydrocortisone, 1% penicillin and streptomycin (P/S), and 100 ng/ml cholera toxin. MCF10DCIS cells were grown in DMEM/F12 media supplemented with 5% horse serum, 1% P/S. Eph4Ras cells were cultured as previously described in MEGM mixed 1:1 with DMEM/F12 media supplemented with 10 ng/ml hEGF, 10 mg/ml insulin, 0.5 mg/ml hydrocortisone, 1% P/S. HEK-293T cell were cultured in DMEM media supplemented with 10% fetal bovine serum (FBS) and 1% P/S. Cells were tested for mycoplasma using the Lonza mycoAlert mycoplasma detection kit. MCF10A and 293 T cells were purchased from the ATCC. MCF10DCIS cells were obtained from the Miller laboratory (Wayne State University, Detroit). Eph4Ras cells were obtained from the Reichmann laboratory (Zurich, Switzerland).

### DNA constructs
The TYK2 cDNA was subcloned into the retroviral pWZL-Blast backbone. FLAG-Tagged constructs and mutants were obtained by site-directed mutagenesis using the Quick-change II kit (Agilent). The Twist1 shRNA constructs in pSP108 lentiviral vector was previously reported[31]. Twist1 shRNA3, 5′-AAGCTGAGCAAGATTCAGACC-3′. Twist1 shRNA5, 5′-AGGTACATCGACTTCCTGTAC-3′. Control shRNA (GFP shRNA), 5′-GCAAGCTGACCCTGAAG-3′. All other shRNA sequences in pLKO.1 lentiviral vector (Sigma-Aldrich) are listed in Supplementary Table 1.

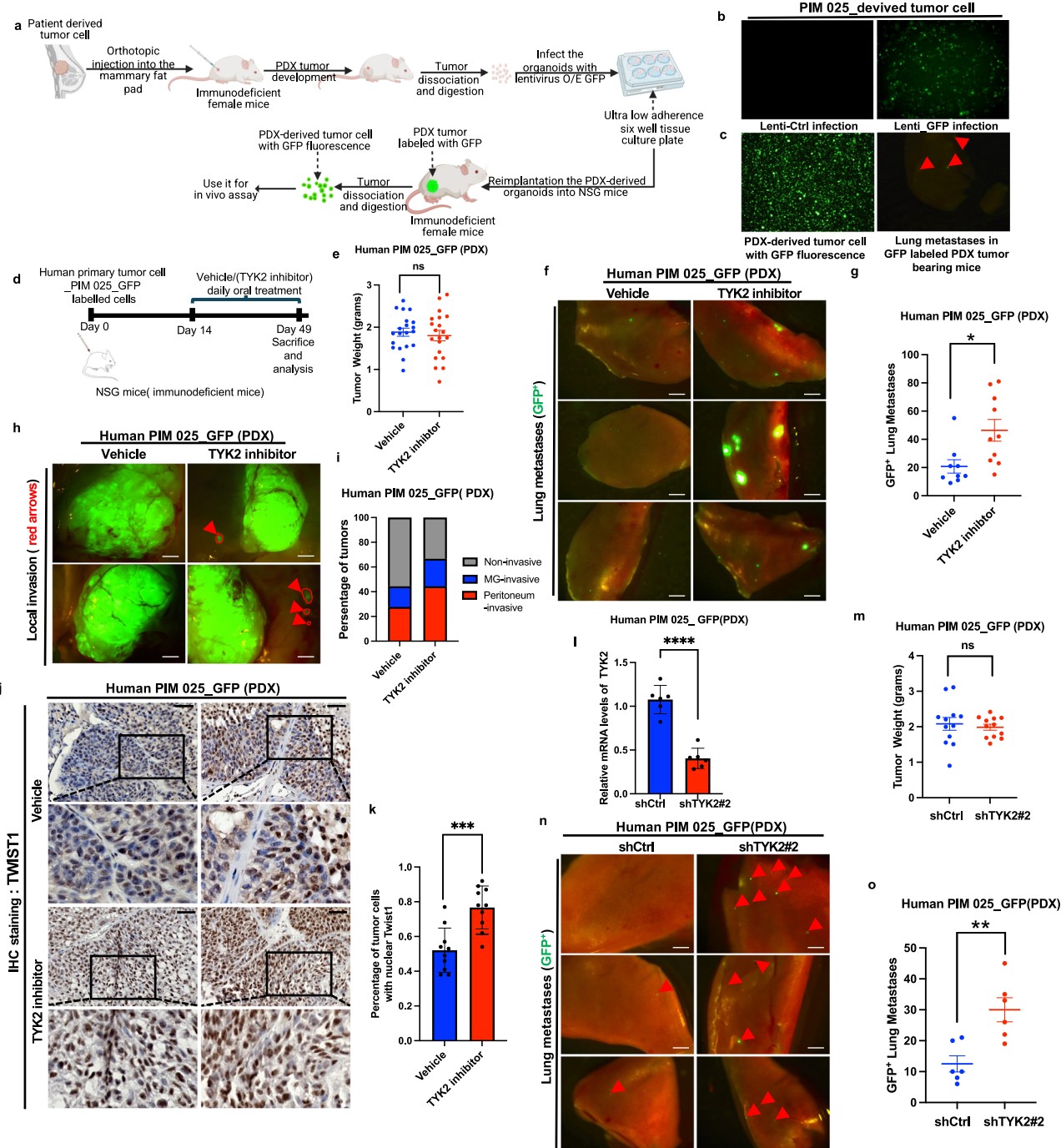

**Fig. 7 | Pharmacologic inhibition or genetic knockdown of TYK2 promotes breast cancer invasion and metastasis in patient-derived breast tumor xenografts. a–c** Schematic illustrating the isolation of patient-derived organoids, GFP labeling, and orthotopic transplantation of GFP-labeled organoids into the mammary fat pad of NSG mice (Created in BioRender. https://BioRender.com/3be9khp) (**a**). Fluorescence microscopy images showing GFP expression in patient-derived organoids (**b**). GFP-labeled patient-derived organoids were orthotopically injected into the mammary fat pad of NSG mice, with GFP-positive tumor cells detected in primary tumors and lungs (**c**). **d–k** Schematic of the TYK2 inhibitor treatment protocol: one million GFP-labeled patient-derived xenograft (PIM025) tumor cells were orthotopically injected into the mammary fat pad of female NSG mice, followed by randomization after two weeks and daily oral gavage with vehicle control or the TYK2 inhibitor deucravacitinib for five weeks before endpoint analyses (**d**). Tumor weight at the experimental endpoint, n = 20 tumor from 10 mice per group (**e**). Representative images of lung metastases, with GFP-positive metastatic tumor cells shown in green, scale bar, 1 mm (**f**), and lung metastatic burden quantified as

the number of GFP-positive nodules (**g**) (*n* = 10 mice per group). Representative images of primary tumor invasion, with arrows indicating GFP-positive invasive tumor cells, scale bar, 2 mm (**h**), and percentage of primary tumors exhibiting local (mammary gland) or regional (peritoneum) invasion(**i**) (*n* = 10 mice per group). Immunohistochemical analysis of TWIST1, scale bar, 50 μm (**j**) and quantification of nuclear TWIST1 as the percentage of total cells (**k**) (*n* = 5 mice per group).**l–o** qPCR analysis of PIM025 tumors expressing control or TYK2-targeting shRNAs, *n* = 3 biological replicates (**l**). Tumor weight at the experimental endpoint, *n* = 12 tumor from 6 mice per group (**m**). Representative images of lung metastases, with arrows indicating GFP-positive metastatic tumor cells, scale bar, 1 mm (**n**), and lung metastatic burden quantified as the number of GFP-positive nodules (**o**) (*n* = 5 mice per group). Data are mean ± SEM, ****$p < 0.0001$; ***$p < 0.001$; **$p < 0.01$; *$p < 0.05$; ns, not significant. Statistical significance was assessed using unpaired two-tailed Student's t-test with Welch's correction. Exact *P* values and source data are provided in the Source Data file.

## Lentivirus and retrovirus packaging and stable cell line construction

Lentivirus and retrovirus were generated by transfecting HEK-293T cells seeded at ~60% confluency in 10 cm dish in complete DMEM, TransIT-LT(Mirus Bio) regents was used for transfection, cells were transfected with 2.3 μg of the pLKO.1(for lentivirus) or pWZL-Blast (for retrovirus) plasmid of interest, 2.07 μg of pCMVD8.2 R (for lentivirus) or pUMVC3 (for retrovirus) and 0.23 μg of VSVG diluted in serum free DMEM. The virus-containing medium was collected and filtered through 0.45 μm sterile filters at 48 h and 72 h after transfection, The lentiviral supernatant was concentrated using Lenti-X™ concentrator. Stable gene knockdown or overexpression cell lines were generated using lentiviral and retroviral. Briefly, shRNA mediated knockdown and overexpression target constructs were introduced by infection with lentiviruses and retrovirus. Concentrated viral supernatants were applied to target cells with 6 μg/ml protamine sulfate to increase the infect efficiency. Infected cells were then selected with 2 μg/ml puromycin or 5 μg/ml blasticidin.

## Three-dimensional (3D) cell culture in polyacrylamide hydrogels of various stiffness

Polyacrylamide hydrogels were prepared as previously described. Briefly,12 mm and 50 mm coverslips were etched using UV/Ozone Procleaner Plus, functionalized using 3-aminopropyltriethoxysilane (Sigma-Aldrich), rinsed with dH$_2$O, incubated in 0.5% glutaraldehyde in PBS for 30 mins, after dried, and then 40% acrylamide/ 2% bis-acrylamide mixtures with different ratio provide stiffness range from 150pa to 5700pa (Atomic Force Microscopy (AFM) were used to measure the stiffness according to Young's modulus of elasticity), 10 μL of 10% ammonium persulfate (APS) and 1 μL of tetramethylethylenediamine (TEMED) (1% and 0.1% of total volume, respectively) were used to induce polymerization, Acr/Bis mixtures polymerized between the functionalized coverslip and a glass slide coated with dichlorodimethylsiloxane. Polyacrylamide-coated coverslips were then washed two times with dH$_2$O, incubated with 1 mM Sulpho-SANPAH dissolve in 20 mM pH 7.5 HEPES buffer under 365 nm ultraviolet light for 10 min, rinsed three times with 50 mM pH 8.5 HEPES buffer, incubated at 37 °C overnight with rat tail Collagen I in PBS, rinsed twice in PBS, and sterilized. The cells were grown in 3D cell culture as previously described[51,52], MCF10A, Eph4Ras and MCF10 DCIS cells were detached with 0.05 % trypsin at 37 °C for 5 ~ 10 min to ensure complete collection of the cells, and then resuspended in resuspension medium (DMEM/F12 supplemented with 20% horse serum and 1% P/S). The cells were washed with assay medium (DMEM/F12 supplemented with 2% horse serum, 5 ng/ml human EGF, 10 mg/ml insulin, 0.5 mg/ml hydrocortisone, 1% P/S)), and then resuspended the cell in assay medium at 4 °C. Seeding the cells in collagen coating stiffness PA-gels, 2% Matrigel should be added to the assay medium, the cells were refed by using assay medium containing 2% Matrigel every 3 days. After 5 days of culture, the acini were harvested for IF staining and western blot analysis.

## Invasive acini quantification

Invasive acini were quantified from brightfield images acquired at low magnification (typically 4X or 10X). For each condition, a minimum of five randomly selected fields of view were analyzed per experiment, and each experiment was repeated in at least three independent biological replicates. Image analysis was performed using ImageJ. Acini (Organoids) were manually counted and categorized into two groups: **Non-invasive acini**, compact spheroids with smooth and continuous borders; **Invasive acini**, structures that exhibited an irregular or spreading morphology, including multicellular protrusions extending outward from the main body or dispersed cells migrating away from the spheroid. Data were normalized to the total number of acini per field, and results are presented as the percentage of invasive acini per condition. Representative images are included alongside the quantification to illustrate morphological criteria.

## 3D confocal microscopy

Our protocol was adapted from the method described by Debnath et al (Debnath et al., 2003). Briefly, after 5 days of 3D culture, acini were wash with 5 mM EDTA in PBS at 4 degree for 30 mins to remove the Matrigel in the system, then acini were fixed with 2% paraformaldehyde (PFA) for 20 minutes at room temperature, permeabilized with 0.5% Triton X-100 in PBS for 10 minutes, quenched with 3 washes of 100 mM PBS-glycine, and then blocked with 20% goat serum in immuno-fluorescence (IF) buffer (1% BSA, 0.05% Tween-20, 7.7 mM NaN3 in PBS) overnight at 4 degree, second blocking was performed for mouse antibodies, we use 7.2% MOM regents with 20% goat serum in IF buffer blocking for 1 h in room temperature. Samples were incubated with primary antibodies ((1:50 to 1:200 dilution depends on the antibody) overnight with 20% goat serum in IF buffer at 4 C, washed 3 times with IF buffer, incubated with secondary antibodies (1:200 dilution) for 1 h at room temperature, and mounted in DAPI-containing mounting medium (Vector Laboratories). Confocal images were acquired using an Olympus FV1000 with 405, 488, 555, and 647 laser lines. Images were linearly analyzed and pseudo colored using ImageJ analysis software.

## Immunoprecipitation and Simple Western blot

3D cultured Cells were incubate with cold PBS-EDTA(5 mM) supplement with protease/phosphatase inhibitors in 4 °C for 30 min, then quick wash with cold PBS, then cells were directly lysed with lysis buffer (20 mM Tris-HCl, 1% Triton X-100, 10 mM MgCl2, 10 mM KCl, 2 mM EDTA, 1 mM NaF, 1 mM sodium orthovanadate, 2.5 mM beta-glycerophosphate, 10% glycerol, with protease inhibitor cocktail (CalBiochem) pH 7.5), scraped off the culture dish, sonicated, supplemented to 400 mM NaCl, sonicated, and diluted to 200 mM NaCl. Lysates were precleared with protein G (for LYN) or protein A (for TYK2) beads for 1 h at 4 °C. Antibodies were conjugated to protein G beads (LYN) or protein A beads (TYK2) (Invitrogen), crosslinked using disuccinimidyl suberate (Pierce) as per manufacturer's protocol, incubated with lysates overnight at 4 °C, washed eight times with IP lysis buffer supplemented with 200 mM NaCl, and eluted by incubation in LDS sample buffer with 50 mM DTT at 70 °C for 10 min. For samples run on Wes (Simple Western), IP samples were eluted by incubation in 1% SDS and 1 mM DTT at 95 °C for 5 min. Dilute lysate as necessary with 0.1X Sample Buffer ((final concentration 0.4 mg/mL-1.0 mg/mL), combine 1-part 5X Fluorescent Master Mix with 4 parts diluted lysate in a microcentrifuge tube, gently mix and boiled at 95 °C for 5 min, store on ice for running WES. Antibodies used in this study included TYK2 specific for human, Sigma-Aldrich, HPA005157; TYK2, GeneTex, GTX61449; pTyr1054/1055 TYK2, Cell Signaling Technology, #68790; Phospho-Tyrosine, Cell Signaling Technology, # 9411; TWIST, Santa Cruz Biotechnology, sc-81417; G3BP2, Sigma-Aldrich, HPA018425; E-cadherin, BD Biosciences,610181;Vimentin, Cell Signaling Technology, #5741; Fibronectin, Sigma-Aldrich, F3548; STAT3, Cell Signaling Technology, #9139; pTyr705 STAT3, Cell Signaling Technology, #9145; STAT1, Cell Signaling Technology, #9172; pTyr701 STAT1, Cell Signaling Technology, #9167; STAT5, Cell Signaling Technology, #57580; pTyr694 STAT5, Cell Signaling Technology, #4322; EPHA2, Cell Signaling Technology, #6997; pS897 EPHA2, Cell Signaling Technology, #6347; LYN, Cell Signaling Technology, #2796; pY416 SFK, Cell Signaling Technology, #2101; IFNAR1, Abcam, ab45172; FLAG, Cell Signaling Technology, #14793; Rabbit IgG, Cell Signaling Technology, #2729; FAK, Cell Signaling Technology, #13009; pTyr397 FAK, Invitrogen, # 44-625 G; AIIB2 β1-integrin-blocking antibody, Developmental Studies Hybridoma Bank, AB_528306; Laminin V, kind gift from M. Aumailley, University of Cologne, Germany; KRT5, Invitrogen, # PA5-

32465; Ki-67, Invitrogen, # MA5-14520; GAPDH, GeneTex, GTX100118.

## RNA extraction and qPCR

RNA was extracted from cells, mice breast tumors using TRI Reagent (Sigma Aldrich) or NucleoSpin RNA II kit (Takara). cDNA was generated using random hexamer primers and a cDNA Reverse Transcription Kit (Applied Biosystems) according to the manufacturer's instructions. Expression values were generated using $\Delta\Delta Ct$ values normalized to GAPDH. Experiments were performed in biological and technical triplicate using 7500 Fast (Applied Biosystems) and CFX Connect (Bio-Rad) real-time PCR detection systems. For data analysis in each comparison (one shRNA vs. the control shRNA), unpaired two-tailed Student's T-tests was used to determine statistical significance. Primers for qPCR are listed in Supplementary Tables 2.

## Proximity ligation assay

Cells were cultured on 3D stiffness polyacrylamide hydrogel for 5 days and fixed and processed as described for immunofluorescence before performing Duolink PLA (Sigma Aldrich) as per manufacturer's protocol. Briefly, mouse anti-TWIST1 and rabbit anti-G3BP2 primary antibodies were used to detect endogenous proteins and subsequently recognized using species specific plus and minus PLA oligonucleotide conjugated probes at 37 °C for 60 min. Interacting probes were then ligated at 37 °C for 30 min and detected by polymerase mediated amplification at 37 °C for 100 min and subsequently analyzed by fluorescent confocal microscopy. For analysis of formed 5-day acini, a minimum 50 cells from 5 random fields were quantified per condition. To quantify the PLA signal, images were converted to 8-bit images and thresholded, the area of PLA signal was then quantified and normalized to cell number using ImageJ.

## Immunohistochemistry (IHC) staining

Paraffin-embedded tumor sections were rehydrated using standard protocols through xylene and graded alcohols. Antigen retrieval was then performed in a pressure cooker (125 °C × 30", 90 °C × 10") in 10 mM Citric acid, pH 6.0, followed by quenching of endogenous peroxidase activity by incubating the samples with 3% H2O2 for 10 min. Samples were blocked in 2.5% Normal Horse Serum Blocking Soultion (Vector Labs) for 2 h and stained with anti-TWIST1 (Santa Cruz, sc-81417, 1:100) overnight at 4 °C. Samples were then incubated with ImmPRESS HRP Horse Anti-Mouse IgG Polymer Detection kit (Vector Labs) at room temperature for 30 min. ImmPACT DAB Substrate Kit (Vector Labs) was used for antigen labeling as indicated by the manufacturer. Stained slides were counterstained with hematoxylin. Images were obtained on a Keyence BZX710 microscope.

## Establishment of PDX-derived organoids in 3D cultures

Patient-derived xenograft models were established by Helen Piwnica-Worms (MD Anderson)[29,30]. PDX-derived organoids (The PIM025 and PIM046 patient-derived organoids were used in this study) were obtained by the adaptation from the following methods[29,53]. Briefly, tumor tissue was harvested, washed in HBSS buffer and dissociated into small fragments around 0.5 mm. Minced tissue was then transfered to a 50 ml Falcon tube containing digestion media (DMEM-F12, 2 mg/ml collagenase A, 2% BSA, 5ug/ml insulin, 0.2% trypsin, 1xAntibiotic/Antimycotic) and incubated at 37 °C in an orbital rotator. Tumors were digested up to 1 h, with monitoring every 20 mins to avoid over digestion of the organoids into single cells. Following enzymatic digestion, PDX-derived organoids were centrifuged for 10 min at 250 × g and treated with 2 ml of warm 5U/ml DNase for 1–2 min until the samples lost viscosity. Following resuspension in culture media (DMEM/F12, 1xAntibiotic/Antimycotic, and 5% bovine calf serum), PDX-derived organoids were separated from single cells (mostly, mouse fibroblasts and blood cells) through five pulsed

centrifugations (< 5 s, 250 × g). PDX-derived organoids were then filtered through a 100um diameter filter mesh to remove any remaining large tumor pieces and resuspended in cold PDX-organoids culture media (DMEM/F12, 1xAntibiotic/Antimycotic, 5% FBS,10 ng/ml hEGF,10 mM HEPES, 1 μg/ml hydrocortisone, 5 μg/ml insulin) with 2% commercial Matrigel (Corning). Organoids were seeded in the polyacrylamide hydrogel for 3D culture as previously described. Brief, Patient-derived xenograft organoids (PDXOs), were embedded within or seeded on top of the collagen-coated PA hydrogels with stiffness range from 150 Pa to 5700 Pa and maintained in complete growth medium at 37 °C with 5% CO$_2$. For PDXO maintenance, we change the PDX-organoids culture media with 2% commercial Matrigel every two days, after 5 days of culture, the organoids were harvested for IF staining and western blot analysis. TNBC PDX-derived organoids presented in this study are organoids freshly isolated from the TNBC PDX tumors and were not passaged in vitro.

## Normal and tumor breast tissue microarrays

Normal human breast tissue slides were provided by Dr. Olu Fadare at UCSD without patient identification information. The breast cancer array BR248a (BIOMAX Inc.) contains 6 cases of normal breast tissues, 17 cases of Invasive carcinoma, and 1 case of medullary carcinoma were stained for TYK2, E-cadherin and DAPI. TMAs were concurrently scanned using the NanoZoomer Slide Scanner with a 10X objective and scored blindly.

## Xenograft tumor assay and PDX-derived tumor assay in vivo

For xenograft tumor assay, GFP-labeled MCF10DCIS cells suspended in serum free DMEM/F12 medium with 50% Matrigel, one million cells were orthotopic injected bilaterally into the inguinal mammary fat pads of 6 to 8-week-old female NSG mice. Tumors were allowed to grow for 2 weeks until they reached 1-3 mm in diameter. Mice were then treated for 3 ~ 5 weeks with daily oral administration of vehicle control (90% PEG300, 5% EtOH, 5% TPGS) or the TYK2 inhibitor deucravacitinib (30 mg/kg/day) purchased from InvivoChem. Mouse weight and tumor sizes were monitored post tumor cell implantation, until mice were sacrificed, and tumor burden analyzed. Mice were dissected and tumor invasion assessed in situ using a fluorescent dissection scope (Leica Microsystems). For PDX-derived metastasis studies, the PIM025 PDX-derived organoids were labeled with GFP using high titter concentrated lentivirus with a pRRL-GFP vector. Lentiviral infection was performed in ultra-low adherence 6-well tissue culture plates overnight, in PDX organoids culture media (DMEM/F12, 1xAntibiotic/Antimycotic, 5% FBS,10 ng/ml hEGF,10 mM HEPES, 1 μg/ml hydrocortisone, 5 μg/ml insulin). 6 μg/ml protamine sulfate was used to increase the infect efficiency. After overnight infection, organoids were washed and re-suspended in 50% Matrigel with serum-free DMEM/F12 media, followed by their orthotopic injection bilaterally into the inguinal mammary fat pads of 6 to 8-week-old female NSG mice. Female mice were used in this study because breast cancer predominantly occurs in females. All work with animals was performed in accordance with UC San Diego IACUC and AAALAC guidelines. The IACUC-approved maximal tumor size limit was 1.5 cm in diameter, tumors were measured daily after reaching 1.0 cm, and this limit was not exceeded in any experiment.

## Statistical analysis

Statistical analyses were performed using GraphPad Prism software. For experiments comparing two groups, statistical significance was assessed using an unpaired, two-tailed Student's $t$-test (with Welch's correction when variance was unequal). For comparisons involving three or more groups, a one-way or two-way ANOVA was performed, followed by Dunnett's multiple comparisons test. Data are presented as mean ± SEM, and $p < 0.05$ was considered statistically significant. Error bars indicate SEM. All qualitative representative data shown were

repeated in at least 3 independent biological replicates. In vivo experiments were designed including multiple mice (n ≥ 6) and repeated in independent biological replicates.

## Ethics statement

All experimental animal studies were conducted under the approval of the Institutional Animal Care and Use Committees (IACUCs) of UCSD Lab Animal Research and were performed in an Association for the Assessment and Accreditation of Laboratory Animal Care-accredited facility.

## Reporting summary

Further information on research design is available in the Nature Portfolio Reporting Summary linked to this article.

## Data availability

All data supporting the findings of this study are available within the paper and its Supplementary Information. Source data are provided with this paper.

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

## Acknowledgements
We thank Dr. Tony Hunter and other members of the Yang lab for technical advice and helpful discussions. We thank the La Jolla Institute microscopy core, in particular Z. Mikulski for SHG imaging. We thank the UCSD Shared Microscope Facility and UCSD Cancer Center Support Grant P30 CA23100 from NCI. We thank Dr. Serge Y. Fuchs at Univ. of Pennsylvania for the human IFNAR1 construct and Dr. Oluwale Fadare at UCSD for human normal breast tissue slides. This work was supported by grants from NCI (RO1CA174869, RO1CA262794, RO1CA268179, and RO1CA236386), the 2021 AACR-Bayer Innovation and Discovery Grant 21-80-44-YANG, The St. Baldrick's Foundation Research Grant, and Krueger v. Wyeth research award to J.Y. PDX models were established through a generous gift from the Cazalot family and from funds from the MD Anderson Cancer Center Breast Cancer Moon Shot Program. Addi-tional funding sources that supported this work include the Cancer Prevention and Research Institute of Texas RP150148 and RP160710 (to H.P.-W.). H.P.-W. is an ACS Research Professor. Z. H. was supported by a Pfizer Oncology-Cell Signaling San Diego Postdoctoral Fellowship. H.S.M. was supported by NIH Pre-doctoral Training grant T32GM007752 and by a predoctoral NRSA fellowship from NCI (F31CA213800). L.F. was supported by an AACR Basic Cancer Research fellowship. A.M.F. was supported by TRDRP Postdoctoral Award T32FT4922.

## Author contributions
Conceptualization: Z.H., H.S.M, and J.Y.; Methodology: Z.H., H.S.M, A.M.F., L.F.; Investigation: Z.H., H.S.M, A.M.F., L.F., S.C. Y.Z. and A.L.; Writing–Original Draft: Z.H., H.S.M., and J.Y.; Writing–Review and Edit-ing: L.F., J.Y., H.P.-W. and D.-E. Z.; Funding Acquisition: Z.H., H.S.M, A.M.F., L.F., J.Y. and H.P.-W; Resources: S.C., S.D., H.P.-W, and K.-I. A.; Supervision: J.Y.

## Competing interests
The authors have no competing interests to declare.
