## [Transparent Peer Review file · Nature Communications]

Extracellular matrix rigidity controls breast cancer metastasis via TYK2-mediated mechanotransduction

Corresponding Author: Professor Jing Yang

Version 0:

Reviewer comments:

Reviewer #1

(Remarks to the Author)

The study by Hu et al represents another chapter by the Yang group to elucidate how mechanotransduction impacts Twist1 activation and nuclear transport. Here the authors show that Tyk2 suppresses EMT programs and TNBC metastasis under low tension ECM conditions, an event that is inactivated when cells are exposed to high tension ECM conditions. Mechanistically, the authors show that Tyk2 dissociates from IFNAR1, resulting in Lyn activation and subsequent phosphorylation of Twist1, leading to its release from G3BP2 complexes. Importantly, genetic inactivation of IFNAR1 phenocopies EMT induction elicited by Tyk2 inhibition. Additionally, the authors show rigidity-driven inactivation of Tyk2 elicits EphA2 stimulation that presumably participates in breast cancer progression and dissemination. Finally, the authors demonstrate that pharmacologic or genetic inactivation of Tyk2 can drive enhanced TNBC metastasis in mice. These findings are rigorous and well-controlled; they also support the major tenets and conclusions of the study. In general, my comments and concerns are relatively minor and provided below under "Specific Comments."

Specific Comments:

- 1) The primary weakness and/or missing component of this intriguing study relates to establishing the underlying mechanism whereby mechanotransduction promotes the dissociation of Tyk2 from IFNAR1. The authors have previously excluded B1 integrins as the point of signal initiation for Twist1 in response to mechanotransduction. However, native collagen I can bind and activate α v integrins, particularly α v β 6 and α v β 8, both of which are expressed in TNBCs and drive disease progression and immune evasion. Likewise, Matrigel components and denatured collagen I can readily activate α v β 3 integrins, which are also expressed on TNBCs and play a prominent role in eliciting EMT programs and metastasis. Thus, some attempt to further explore integrin involvement appears warranted.
- 2) Along these lines, the authors previously showed that ligand-independent activation of EphA2 figured prominently in the activation of Twist1 by Lyn. Here the authors show that the binding of Tyk2 to either IFNAR1 or EphA2 occurs in soft ECM microenvironments, but not in rigid ECM microenvironments. Given this finding, it is unclear why the authors failed to probe Tyk2 immunocomplexes for EphA2 in Figure 5n. This should be shown (e.g., immunoblot or immunofluorescence), as perhaps increased ECM rigidity drives ligand independent activation of EphA2, which serves to uncouple the formation of a tripartite complex comprised of IFNAR1:Tyk2:EphA2 that is bridged by Tyk2.
- 3) Additionally, the findings point to post-translational regulation of IFNAR1 and EphA2 expression, such that ECM rigidity promotes degradation of IFNAR1 and stabilization of EphA2. What accounts for this inverse relationship and how do these events underlie inactivation of Tyk2 and the subsequent induction of EMT programs.
- 4) Throughout the paper, the authors present impressive alterations in the size and morphology of TNBC organoids. These changes go beyond those associated with simple induction of EMT programs and likely reflect a strong proliferative component in rigid ECM. As such, some measurements of cellular proliferation or lack thereof needs to be presented.
- 5) To what extent are these events reversible? Can cells post-Tyk2 inactivated cells cultured in stiff microenvironments undergo MET programs when returned to soft microenvironments? This is an important question to address both in 3D-cultures and pulmonary metastases in mice - i.e., lung elasticity is dramatically different than that of primary tumors, so what

is the status of Tyk2 and EMT programs in pulmonary metastases of mice treated with Tyk2 inhibitors?

6) Finally, essentially everything metastases in NSG mice, and as such, it would be interesting to determine whether Tyk2 inhibition is sufficient to drive TNBC metastasis in a syngeneic mouse model (e.g., Eph4Ras cells) to assess the impact of Tyk2 inhibition in a more stringent and immunocompetent model.

Reviewer #2

(Remarks to the Author)

Hu et al. investigated the role of TYK2 in regulating breast cancer cell invasiveness in dependence on extracellular matrix mechanical cues. The authors provide evidence for a novel cell-autonomous role of TYK2 in inhibiting breast cancer cell invasiveness. Using TYK2 knockdown and pharmacological inhibition of TYK2 in human and murine breast cancer 3D models and patient-derived PDX models, authors show that TYK2 blocks breast cancer cell invasiveness at low ECM stiffness. Mechanistically, authors show that TYK2 blocks the phosphorylation of EphA2 and Lyn, thereby preventing nuclear translocation of TWIST1, which has been shown to promote cell invasiveness in a previous study from the group. Authors furthermore show that this function of TYK2 depends on the presence of the type I interferon receptor 1 (IFNAR1) but not on classical IFN signalling, as neither the knockdown of IFNAR2 nor the pharmacological inhibition of JAK1 or STAT3/STAT5 affects cell invasiveness. With respect to mechanical cues, authors show that transitioning from low to high ECM stiffness, TYK2 re-localizes from the cell membrane to the cytoplasm and fails to block cell invasiveness. Overall, this is an interesting and thoroughly performed study that adds a new facet to TYK2 biology. The study is highly relevant, especially when considering the growing use of TYK2 inhibitors in the clinics. However, some concerns should be addressed to improve the manuscript.

Major comments:

- 1) The title should be modified. Coining the term “mechanosensing kinase TYK2” is to my opinion not supported by the data. Authors show that TYK2 relocates to the cytoplasm at high ECM stiffness but do not provide data on the protein levels and the localization of IFNAR1. It is possible that membrane rigidity controls IFNAR1 surface levels and that this then effects the localisation of TYK2. It is also possible that high ECM rigidity induces conformational changes in IFNAR1 that prevent TYK2 binding. Whether or not TYK2 is actually “sensing” differences in membrane rigidity and alternative possibilities should be discussed.
- 3) Some controls are missing: (i) Authors should provide proof that the pharmacological inhibition of STAT3 and STAT3/STAT5 was effective in their experimental system. (ii) Authors should provide proof that the knockdown of IFNAR2 was efficient in their experimental system. They show significant reduction of *Ifnar2* mRNA but not whether this was sufficient to abolish IFN-I signalling. Authors should e.g. perform the same control experiments as shown for knockdown of IFNAR1 (Extended data Fig. 4j).
- 4) Statistical analysis: Authors report that all p values were derived from Student's t-test with Welch correction. However, they mostly have 3 or more treatments to compare in their experimental settings. ANOVA should be used instead of pairwise comparisons with t-test.
- 5) For all Western blot data, authors should provide information on the sample sizes (representatives of x experiments or only done once?)
- 6) The manuscript would greatly profit from thorough editing. There are a large number of typos, grammatical errors and long sentences that are difficult to understand.

Minor comments:

- 1) Introduction, page 3: “....activates downstream STAT proteins upon cytokine-stimulated receptor dimerization” should be corrected to “....activates downstream STAT proteins upon cytokine-stimulated conformational changes of receptor complexes”.
- 2) Results, page 5: “.....Consistent with the human TYK2 results, knockdown of mouse TYK2 led to a high percentage of mammary organoids undergoing EMT with corresponding changes in EMT markers and invasion at low stiffnesses (Fig. 1h,i and Extended Data Fig. 1a)”. There are no data on EMT markers shown in the Figures. Please correct.
- 3) Results, page 10: “.....involves interferon-induced heterodimerization between IFNAR1 (a direct binder of TYK2) and IFNAR2 (a direct binder of JAK1).....”. Correct to “.....involves heterodimerization between IFNAR1 (a direct binder of TYK2) and IFNAR2 (a direct binder of JAK1).....”.
- 4) Figure 3j: TYK2 levels strongly increase with increasing ECM stiffness in the sh-control, which is not commented by the authors and seems to contradict later results (e.g. Fig. 4a, Ext Data Figure 1a). Please clarify.
- 5) Figure 4f. Change figure caption to STAT3/5 inhibitor (instead of pan-STAT inhibitor).
- 6) Results page 12/13: “Deucravacitinib treatment reduced phospho-STAT3 signal in primary tumors, indicating effective inhibition of the TYK2 kinase activity (Supplementary n. Fig. 6e).” Supplementary Figure 6e (i.e. Extended Data Figure 6e) shows something different. P-STAT3 is shown in Extended Data Figure 5d. Please correct.
- 7) Discussion, page 14: “At high stiffnesses of breast tumor tissues, TYK2 dissociates with IFNAR1 from the cell membrane,”. It remained unclear whether IFNAR1 dissociates with TYK2 from the membrane or not. Also see major comment 1. Different possibilities should be discussed (see also major comment 1).

Reviewer #3

(Remarks to the Author)

This study by Hu and Majeski et al. presents an extensive investigation into the role of TYK2 as a mechanosensitive suppressor of EMT and invasion in breast epithelial and tumor cells. The authors propose a noncanonical pathway in which soft ECM conditions maintain TYK2 at the membrane to repress TWIST1 nuclear localization via EPHA2 and LYN, independently of classical JAK–STAT signaling. The work includes stiffness-controlled 3D cultures, PDX-derived organoids, in vivo xenografts, and patient tissue analysis. The study is interesting, proposes a new mechanism by which tissue mechanics can regulate invasive behavior and metastasis, and is particularly relevant given the therapeutic use of TYK2 inhibitors to modulate immune responses in autoimmune disease. However, key mechanistic steps are dissected only in non-tumorigenic models and are not validated in PDX-derived organoids. Invasion quantification lacks cellular resolution, and the role of canonical mechanotransducers is not addressed. In addition, conclusions regarding the signaling axis and basal identity require stronger experimental support. Overall, this is a compelling study, but major revisions are required before publication in Nature Communications. Some of my detailed comments are summarized below.

Major Comments

1. Mechanistic validation in clinically relevant models is insufficient. The proposed TYK2-EPHA2-LYN-TWIST1 axis is dissected entirely in MCF10A and Eph4Ras models but generalized to patient-derived TNBC. In PDX-derived organoids, only invasion phenotypes are tested. In the in vivo analysis using PDXOs (e.g., Fig. 7j), the TWIST1 nuclear signal differences are not clear; quantitative analysis (nuclear/cytoplasmic ratios) and nuclear counterstaining are needed. Additionally, mechanistic validation in PDXOs, such as showing changes in EPHA2 phosphorylation and TYK2 localization, is essential to support the claim of a conserved metastasis suppressor mechanism in human tumors.
2. Definition and quantification of invasion lack resolution. Invasion is inferred from morphological features (e.g., protrusions, spreading) in brightfield images, which is insufficient to distinguish true matrix invasion from acinar flattening or spreading. Higher magnification images or confocal z-stacks are needed. In spread conditions, for instance, in Figs. 1h, 1l, 2i, and S1e, overlapping acini/organoids complicate quantification. The authors should clarify how invasive fractions were determined, especially under these conditions.
3. Canonical mechanotransducers are not addressed. The study does not investigate integrin-mediated mechanotransduction, including integrin expression, FAK/SRC activation, or YAP localization, which are key regulators of stiffness-driven invasion. Given the potential convergence between EPHA2, TYK2, LYN, and integrins, this pathway should be considered—for example, by assessing focal adhesion signaling or YAP localization following TYK2 inhibition. Moreover, the role of the ECM warrants deeper discussion. Although TYK2 delocalization at high stiffness is described, the upstream mechanical inputs regulating this process remain undefined. Future work should explore whether force-sensitive mechanisms—such as integrin signaling, cytoskeletal tension, or membrane compartmentalization—govern TYK2 localization and activity, particularly in physiologically relevant, collagen-rich 3D environments. This represents a significant gap in understanding how TYK2 regulates invasion in a mechanically dynamic context.
4. ECM composition does not model physiological invasion contexts. The stiffness-controlled ECM model lacks fibrillar collagen I, a key component of breast tumor stroma typically invaded by cancer cells. Coated PA gels with non-fibrillar collagen I do not recapitulate 3D ECM architecture. The role of TYK2 in fibrillar collagen-rich matrices should be tested, and these limitations clearly acknowledged. The authors may also consider testing the effect of TYK2 knockdown on invasion using primary tumors isolated from their in vivo models.
5. JAK-STAT signaling independence is not conclusively shown. The authors show that JAK/STAT inhibition does not phenocopy TYK2 loss but do not assess whether TYK2 knockdown alters JAK1/2 or STAT3/5 phosphorylation. Compensatory effects could influence EMT independently of canonical transcriptional activity. This should be evaluated under varying stiffness conditions.
6. Experimental rationale is unclear in several cases. For example: Why is immunoprecipitation used to detect phospho-TYK2 instead of standard western blot? What is the rationale for using TYK2 overexpression and PLA assays?
7. Incomplete characterization of invasion in vivo. In Fig. 6i, red arrows indicate putative invasive cells, but these may represent sectioning artifacts. This panel is not described in the Results section. More broadly, it is unclear how invasion and dissemination are distinguished from local displacement.
8. Effects on proliferation not assessed. In Fig. 1i, TYK2 knockdown organoids appear larger, suggesting altered growth. This is not tested in PDXOs. Similarly, for in vivo tumors (Fig. 6b, f), tumor weight alone is insufficient to assess growth effects. Additional metrics such as tumor volume and proliferation markers (e.g., Ki-67) should be included. Time-course growth analyses in vitro are also recommended.
9. Terminology and labeling inaccuracies. MCF10A-based acini are referred to as “human basal mammary organoids.” This is misleading; these are not true organoids. Text and figure labels should be revised accordingly.
10. TYK2 overexpression vs. endogenous expression not clearly distinguished. In localization experiments, the authors must clarify whether TYK2 is overexpressed or endogenous. Unless stated otherwise, such experiments should rely on endogenous TYK2.
11. Mechanistic assertions lack critical controls.

- i) The EPHA2–S897 mechanism would be strengthened by using a phosphomimetic EPHA2-S897E mutant to test sufficiency for EMT and invasion.
- ii). Co-localization of TYK2, IFNAR1 and EPHA2 (and phospho-EPHA2) is not shown; this is required to support membrane-dependent regulation as the authors claim that: “We show that at low matrix stiffness of normal mammary glands, TYK2 localizes on the cell membrane by binding to IFNAR1 and associates with EPHA2, thus blocking EPHA2 S897 phosphorylation. At high stiffnesses of breast tumor tissues, TYK2 dissociates with IFNAR1 from the cell membrane, thus allowing EPHA2 S897 phosphorylation and downstream activation of the TWIST1-driven EMT and invasion.”
- iii) In several westerns (e.g., Fig. 3j–k), total EPHA2 appears increased at high stiffness or upon TYK2 inhibition. These changes should be quantified and independently replicated to distinguish phosphorylation from total protein level changes. Similarly, TYK2 signal intensity appears reduced at high stiffness (e.g., Fig. S3b, Fig. 4b); the authors must distinguish between protein relocalization and downregulation.

12. Some claims are not supported by the data shown. For example, on page 5 the authors state: “Knockdown of TYK2 potentially induced EMT, as shown by increased Vimentin and decreased E-cadherin.” However, these markers are not shown. The cited figures quantify invasive acini rather than protein expression. If this conclusion is based on previous literature, this should be explicitly stated.

13. TYK2 function not linked to basal cell identity. The term “basal-subtype organoids” is used without showing basal marker expression. No data are provided on KRT14, KRT5, or p63 in MCF10A or PDXOs. The authors should validate epithelial cell identity and assess whether TYK2 function correlates with basal marker expression. Is TYK2 differentially expressed or localized in invasive cells compared to cells within the acinar body?

Minor Comments

1. Clarify drug concentrations used in Fig. 1b–c and their relevance.
2. Improve figure referencing and ordering; panel arrangement (horizontal vs. vertical) and citation sequence (e.g., S2b before S1a) are inconsistent.
3. Quantification of PIM046 invasion (Fig. S1f) is not referenced in the text.
4. Emphasize and clarify the differences between MCF10A (non-tumorigenic) and tumor-derived models in the main text. Related to that, consider including a statement in the Discussion addressing the potential risk that TYK2 inhibition might promote invasive behavior, and possibly metastasis, in otherwise non-transformed epithelial tissues.
5. Show knockdown efficiency for IFNAR1, IFNAR2, and TYK2 in all relevant models.
6. Validate the efficacy of IFNAR1-blocking antibodies.
7. Some immunostainings are difficult to interpret (e.g., E-cadherin in Fig. 1f). Fibronectin appears increased in invasive conditions, but comparing monolayers to acini may confound interpretation due to differences in antibody penetration. The authors should include single-cell controls or validate penetration, as done in Fig. S3d.
8. Include time-lapse imaging or t = 0 frames for invasion assays, where feasible.
9. Some mechanistic conclusions rely on cited literature rather than directly shown data.
10. Include methodology for PDX organoid culture and maintenance.
11. Is the observed effect on TYK2 localization and activity upon increasing stiffness specific to PA gels coated with collagen I?

Version 1:

Reviewer comments:

Reviewer #1

(Remarks to the Author)

The authors have addressed my previous concerns.

Reviewer #2

(Remarks to the Author)

The authors have thoroughly and sufficiently addressed all my concerns and comments.

Reviewer #3

(Remarks to the Author)

The authors have adequately addressed most of my previous comments. This is a very nice and well-executed manuscript. A few minor points remain:

1. The explanation that TYK2 immunofluorescence is not feasible due to antibody absorption by Matrigel is not fully convincing. Organoids can be efficiently and rapidly (few seconds) recovered from Matrigel using cold media, followed by fixation and staining in suspension or embedding. This approach is routinely used for organoid immunostaining and would allow visualization of TYK2 localization without matrix interference.
2. The statement that YAP does not control TWIST1 subcellular localization seems too narrow and insufficiently supported

by data. YAP and TWIST1 are known to regulate each other's expression and activity in multiple systems, including breast cancer. Even if YAP does not directly influence TWIST1 localization, it could modulate TWIST1 levels through transcriptional and mechanotransduction pathways; which could affect TWIST distribution. The authors should clarify whether YAP activity affects TWIST1 localization or expression in their system, or discuss YAP's potential role in the Discussion. This point is particularly relevant given that the pathway is dependent on β 1-integrin.

3. The new integrin interference data clearly indicate that β 1-integrin engagement is required for stiffness-dependent TYK2 inactivation and EMT induction, whereas FAK activity is dispensable for this step. This mechanistic conclusion should be explicitly stated in the manuscript to ensure alignment between the data and interpretation.

4. The authors' response clarifies that the study focuses on how ECM rigidity enables basement membrane breaching during EMT. However, this is not clearly articulated in the current manuscript. The authors are invited to explicitly define basement membrane breaching as the central biological process under investigation to strengthen conceptual focus and clarify the relevance of the Matrigel system.

5. The architecture and rationale of the experimental setup should be described more clearly. From the response, acini are embedded in Matrigel, surrounded by a self-deposited basement membrane, and exposed to stiffness cues via an underlying collagen I-coated PA gel. This setup models epithelial cells breaching the basement membrane in response to external rigidity. The authors should make this organization explicit, clarifying that (1) acini possess an intact basement membrane, (2) PA gel stiffness represents tissue rigidity outside the basement membrane, and (3) collagen I functionalization enables mechanical coupling across the interface. This explanation would justify the model design and the use of collagen I-coated PA gels.

• Typo: "EMC" → "ECM"

Version 2:

Reviewer comments:

Reviewer #3

(Remarks to the Author)

All my concerns have been addressed, great manuscript.

We thank the reviewers and the editor for their constructive critiques and have performed a large number of new experiments/analyses to address them. The point-by-point responses to individual comments below include detailed information to address all the critical points. We are grateful for all your insightful suggestions and believe that the revision has substantially strengthened this study.

REVIEWER COMMENTS

Reviewer #1 (Remarks to the Author):

The study by Hu et al represents another chapter by the Yang group to elucidate how mechanotransduction impacts Twist1 activation and nuclear transport. Here the authors show that Tyk2 suppresses EMT programs and TNBC metastasis under low tension ECM conditions, an event that is inactivated when cells are exposed to high tension ECM conditions. Mechanistically, the authors show that Tyk2 dissociates from IFNAR1, resulting in Lyn activation and subsequent phosphorylation of Twist1, leading to its release from G3BP2 complexes. Importantly, genetic inactivation of IFNAR1 phenocopies EMT induction elicited by Tyk2 inhibition. Additionally, the authors show rigidity-driven inactivation of Tyk2 elicits EphA2 stimulation that presumably participates in breast cancer progression and dissemination. Finally, the authors demonstrate that pharmacologic or genetic inactivation of Tyk2 can drive enhanced TNBC metastasis in mice. These findings are rigorous and well-controlled; they also support the major tenets and conclusions of the study. In general, my comments and concerns are relatively minor and provided below under "Specific Comments."

Reply: We sincerely thank the reviewer for the careful evaluation of our manuscript and the constructive feedback. We are pleased that the reviewer finds our study rigorous, well-controlled, and supportive of the major conclusions regarding the role of TYK2/IFNAR1 mechanotransduction in regulating Twist1 activation and TNBC metastasis. We have carefully considered individual comments, performed new experiments, and provided detailed responses below. Where appropriate, we have revised the text and figures in the manuscript to address these points. We believe that these changes have further clarified the mechanistic framework of TYK2 regulation by matrix stiffness and strengthened the key conclusions of the study.

Specific Comments:

1) The primary weakness and/or missing component of this intriguing study relates to establishing the underlying mechanism whereby mechanotransduction promotes the dissociation of Tyk2 from IFNAR1. The authors have previously excluded B1 integrins as the point of signal initiation for Twist1 in response to mechanotransduction. However, native collagen I can bind and activate αv integrins, particularly $\alpha v\beta 6$ and $\alpha v\beta 8$, both of which are expressed in TNBCs and drive disease progression and immune evasion. Likewise, Matrigel components and denatured collagen I can readily activate $\alpha v\beta 3$ integrins, which are also expressed on TNBCs and play a prominent role in eliciting EMT programs and metastasis. Thus, some attempt to further explore integrin involvement appears warranted.

Reply: We thank the reviewer for this insightful comment. Indeed, we have been working to understand the involvement of various integrins in activating TWIST1 mechanotransduction ever since we started this project 15 years ago. In Wei et al, NCB 2015 and Fattet et al, Developmental Cell, 2020, we presented data that treatment with integrin beta1 blocking antibody could completely block TWIST1 nuclear localization and EMT at high ECM rigidities. However, expression of activating mutants of ITGB1, such as the V737N clustering mutant that was shown to mimic mechanically activated $\beta 1$ integrin by the group of Dr. Valerie Weaver did not impact TWIST1 localization or EMT/invasion at low or high stiffnesses.

Following the reviewer's advice, we tested a high affinity and selective inhibitor of $\alpha v\beta 6$ GSK3008348 and found no effect on cell invasion at all ECM rigidities, suggesting that blocking integrin $\alpha v\beta 6$ alone is not sufficient to block this mechanotransduction pathway. We also tested GLPG-0187, a nanomolar affinity inhibitor of RGD (Arg-Gly-Asp) integrin receptors ($\alpha v\beta 1$, $\alpha v\beta 3$, $\alpha v\beta 5$, $\alpha v\beta 6$, $\alpha v\beta 8$ and $\alpha 5\beta 1$) and found that GLPG-0187 shows very mild suppressive effect (~20%) on invasion at high rigidities. In contrast, consistent with what we reported in Wei et al., 2015 NCB, the anti-integrin $\beta 1$ blocking antibody AIB2 that blocks cell attachment to fibronectin, collagen-I, collagen-IV, and laminin (Hall et al, JCB, 1990) could completely suppress invasion in all rigidities. It is important to note that all three inhibitors potently reduced FAK kinase activation. Furthermore, inhibiting FAK or Src did not block TWIST1 nuclear localization at high ECM rigidities

(please see data presented for Response to Review 3 Major Concern #3). Based on these results and other published studies, our conclusion is that integrin- $\beta 1$ is essential for cells to bind to ECM and sense ECM rigidities; however, integrin activation alone does not activate the TWIST1 mechanotransduction pathway.

2) Along these lines, the authors previously showed that ligand-independent activation of EphA2 figured prominently in the activation of Twist1 by Lyn. Here the authors show that the binding of Tyk2 binding to either IFNAR1 or EphA2 occurs in soft ECM microenvironments, but not in rigid ECM microenvironments. Given this finding, it is unclear why the authors failed to probe Tyk2 immunocomplexes for EphA2 in Figure 5n. This should be shown (e.g., immunoblot or immunofluorescence), as perhaps increased ECM rigidity drives ligand independent activation of EphA2, which serves to uncouple the formation of a tripartite complex comprised of IFNAR1:Tyk2:EphA2 that is bridged by Tyk2.

Reply: We thank the reviewer for this insightful comment. To address this point, we performed additional co-immunoprecipitation experiments in acini cultured under soft versus stiff ECM conditions, immunoprecipitating TYK2 and probing for EphA2 and IFNAR1. As **New Figure 5m** shows, IFNAR1 knockdown markedly reduced the association of TYK2 with EPHA2, supporting our model that IFNAR1 anchors TYK2 at the plasma membrane to enable its interaction with EPHA2 at low ECM stiffnesses.

Response Fig. 2 (New Fig. 5m): IFNAR1 are required for TYK2 activation and binding with EPHA2. MCF10A acini overexpressing FLAG-TYK2 WT and shRNAs against IFNAR1 or a control shRNA were grown on 3D-PA gels for 5 days. Lysates were subjected to anti-FLAG immunoprecipitation were analysed with immunoblotting.

3) Additionally, the findings point to post-translational regulation of IFNAR1 and EphA2 expression, such that ECM rigidity promotes degradation of IFNAR1 and stabilization of EphA2. What accounts for this inverse relationship and how do these events underlie inactivation of Tyk2 and the subsequent induction of EMT programs.

Reply: In Fattet et al., Developmental Cell, 2020, we reported that Ser897 phosphorylation on EPHA2 is essential for driving TWIST1 nuclear localization and EMT at high ECM stiffness. **New Fig. 5l** shows that IFNAR1 knockdown led to precocious S897 phosphorylation on EphA2 at low ECM stiffness, suggesting that IFNAR1/TYK2 interaction with EphA2 inhibits S897 phosphorylation on EphA2.

Response Fig. 3 (New Fig. 5l) : MCF10A organoids overexpressing FLAG-TYK2 WT and shRNAs against IFNAR1 or a control shRNA were grown on 3D-PA gels for 5 days. Lysates were subjected to anti-FLAG immunoprecipitation and immunoblotted as indicated.

In terms of IFNAR1, our data indeed suggests that ECM rigidity regulates the IFNAR1 protein level. IFNAR1 protein level is reduced at high ECM rigidities (**New Fig. 5l**), which could account for the release of TYK2 from cell membrane. Our preliminary analysis suggests that this change is distinct from the known mechanism of ligand-induced IFNAR1 endocytosis and subsequent lysosomal degradation, therefore extensive future studies will be pursued to open this next chapter.

4) Throughout the paper, the authors present impressive alterations in the size and morphology of TNBC organoids. These changes go beyond those associated with simple induction of EMT programs and likely reflect a strong proliferative component in rigid ECM. As such, some measurements of cellular proliferation or lack thereof need to be presented.

Reply: Mammary acini form a sphere at low ECM stiffnesses, therefore the bright-field and confocal images only represent a partial area or cross-section of the acini sphere. At high stiffnesses, cells invade as a flat sheet and the confocal images capture the entire cell population. Therefore, such images cannot be compared directly to determine total cell number. To further address this question, we performed Ki67 staining. As shown in **New Extended Data Fig.1c,d**, Consistent with published studies from the Valerie Weaver's group, high stiffness increases cell proliferation from 73% to 94%. Interestingly, TYK2 knockdown mildly increased Ki67+ population from 73% to 80% at 320Pa and did not increase Ki67+ population at 5700Pa.

In MCF10DCIS and PIM025 PDX tumor xenografts, we also performed IHC staining for Ki67 in the primary tumor. As shown in **New Extended Data Fig.7d, e** and **New Extended Data Fig.8 g, h**, TYK2 inhibition did not significantly alter Ki67+ tumor cell population in vivo. Taken together, these results indicate that TYK2 blockade does not exert a strong effect on cell proliferation either in vitro or in vivo.

Response Fig. 6 (New Extended Data Fig. 8 g,h): a) Immunohistochemistry staining analysis of Ki67 in Patient Derived Xenograft primary tumors (PIM025) from vehicle and TYK2 inhibitor deucravacitinib treated groups. Scale bar represents 50 μ m. b) Nuclear Ki67 staining of vehicle and TYK2 inhibitor deucravacitinib treated Patient Derived Xenograft primary tumors (PIM025) were quantified and represented as percentages of total cells. Dots represent individual fields. ns, not significant.

5) To what extent are these events reversible? Can cells post-Tyk2 inactivated cells cultured in stiff microenvironments undergo MET programs when returned to soft microenvironments? This is an important question to address both in 3D-cultures and pulmonary metastases in mice - i.e., lung elasticity is dramatically different than that of primary tumors, so what is the status of Tyk2 and EMT programs in pulmonary metastases of mice treated with Tyk2 inhibitors?

Reply: We thank the reviewer for this insightful comment. Indeed, ECM stiffness regulates epithelial-mesenchymal plasticity in a reversible fashion. MCF10 cells are cultured on hard plastic surfaces (>2GPa) where TYK2 is inactivated and TWIST1 is in the nucleus before seeding onto the 3D hydrogel. 24 hours upon seeding on soft rigidities, TYK2 regains membrane localization and TWIST1 exits from the nuclei. Normal lung tissues are around 100-200Pa, similar to normal mammary tissues. However, when metastatic lesions grow in the lung, like primary tumors developing in mammary tissues, metastatic tumor cells recruit stroma cells and remodel ECM to stiffen metastatic lesions in the lung. The current mouse metastasis models present sporadic metastases in the lung and do not allow to easily capture rare early lesion expansions to study metastatic dissemination from the secondary lung lesions. Future studies are worth pursuing to follow both primary and secondary lesion development in a tractable mouse breast tumor metastasis model.

6) Finally, essentially everything metastases in NSG mice, and as such, it would be interesting to determine whether Tyk2 inhibition is sufficient to drive TNBC metastasis in a syngeneic mouse model (e.g., Eph4Ras cells) to assess the impact of Tyk2 inhibition in a more stringent and immunocompetent model.

Reply: We completely agree with the reviewer that the immune system could play critical roles in suppressing tumor development and metastasis of immunogenic tumors. TYK2 is a critical component of the JAK/STAT signaling pathway in various immune cells, including T cells, therefore patients treated with TYK2 inhibitors are also immunosuppressed. In fact, deucravacitinib carries a cancer risk warning due to the known role of TYK2 in immunosuppression. In immunocompetent mice, TYK2 inhibition suppresses mouse immune cell activation, thus contributing to accelerated tumor development and metastasis of immunogenic tumors. Therefore, applying TYK2 inhibitors in immunocompetent mice could increase metastasis via immune suppression and via its intrinsic EMT suppression we discovered, which complicates the interpretation of such experiments. Our results using NSG immunocompromised mice show that even without immune surveillance, human TNBC PDX tumors rarely metastasize, and TYK2 suppression in tumor cells (not immune cells) could significantly increase dissemination, thus demonstrating the intrinsic metastasis suppression function of TYK2 in tumor cells.

Reviewer #2 (Remarks to the Author):

Hu et al. investigated the role of TYK2 in regulating breast cancer cell invasiveness in dependence on extracellular matrix mechanical cues. The authors provide evidence for a novel cell-autonomous role of TYK2 in inhibiting breast cancer cell invasiveness. Using TYK2 knockdown and pharmacological inhibition of TYK2 in human and murine breast cancer 3D models and patient-derived PDX models, authors show that TYK2 blocks breast cancer cell invasiveness at low ECM stiffness. Mechanistically, authors show that TYK2 blocks the phosphorylation of EphA2 and Lyn, thereby preventing nuclear translocation of TWIST1, which has been shown to promote cell invasiveness in a previous study from the group. Authors furthermore show that this function of TYK2 depends on the presence of the type I interferon receptor 1 (IFNAR1) but not on classical IFN signalling, as neither the knockdown of IFNAR2 nor the pharmacological inhibition of JAK1 or STAT3/STAT5 affects cell invasiveness. With respect to mechanical cues, authors show that transitioning from low to high ECM stiffness, TYK2 re-localizes from the cell membrane to the cytoplasm and fails to block cell invasiveness. Overall, this is an interesting and thoroughly performed study that adds a new facet to TYK2 biology. The study is highly relevant, especially when considering the growing use of TYK2 inhibitors in the clinics. However, some concerns should be addressed to improve the manuscript.

Reply: We sincerely thank the reviewer for the thoughtful and constructive evaluation of our manuscript. We are encouraged that the reviewer finds our study interesting, thorough, and relevant to the growing clinical use of TYK2 inhibitors. We appreciate the recognition of our mechanistic findings, specifically the identification of a novel, cell-autonomous role of TYK2 in suppressing breast cancer cell invasiveness in response to ECM stiffness. We have thoroughly revised the manuscript to address the reviewer's concerns and believe that these changes improve the clarity and robustness of our conclusions.

Major comments:

1) The title should be modified. Coining the term “mechanosensing kinase TYK2” is to my opinion not supported by the data. Authors show that TYK2 relocates to the cytoplasm at high ECM stiffness but do not provide data on the protein levels and the localization of IFNAR1. It is possible that membrane rigidity controls IFNAR1 surface levels and that this then effects the localisation of TYK2. It is also possible that high ECM rigidity induces conformational changes in IFNAR1 that prevent TYK2 binding. Whether or not TYK2 is actually “sensing” differences in membrane rigidity and alternative possibilities should be discussed.

Reply: We thank the reviewer for this insightful suggestion. We agree with this suggestion and have modified the title to “via TYK2 mechanotransduction”. We also discussed this point in response to Reviewer #1 Point #3. Indeed, our very preliminary data suggest a change of IFNAR1 protein level at high ECM rigidities, which open a new chapter in our goal to understand how cells convert a mechanical cue from ECM to a defined transcriptional response to drive EMT and invasion. We also added additional discussion in the manuscript Page 16 Paragraph 2.

3) Some controls are missing: (i) Authors should provide proof that the pharmacological inhibition of STAT3 and STAT3/STAT5 was effective in their experimental system. (ii) Authors should provide proof that the knockdown of IFNAR2 was efficient in their experimental system. They show significant reduction of *Ifnar2* mRNA but not whether this was sufficient to abolish IFN-I signalling. Authors should e.g. perform the same control experiments as shown for knockdown of IFNAR1 (Extended data Fig. 4j).

We thank the reviewer for the helpful comment to improve the quality of the data. These additional controls have been added to strengthen our conclusions.

1. For the STAT3 inhibitor C188-0 and the STAT3/5 inhibitor SH-4-54, we examined pSTA3 and pSTAT5 status and found that indeed both inhibitors inhibited p-STAT3 and/or p-STAT5 induced by IFN α/β treatment. These data are now included in **New Extended Data Fig.6e, f**.
2. To confirm the functional consequence of IFNAR2 knockdown, we assessed IFN-I–induced STAT1/3 phosphorylation in IFNAR2-depleted cells. Our results show that IFNAR2 knockdown markedly reduced IFN α/β induced phosphorylation of STAT1 and STAT3, indicating effective disruption of IFN-I signaling. These data are now included in **New Extended Data Fig. 6n**, and the experimental approach mirrors the validation of IFNAR1 knockdown in New Extended Data Fig. 6i.

4) Statistical analysis: Authors report that all p values were derived from Student's t-test with Welch correction. However, they mostly have 3 or more treatments to compare in their experimental settings. ANOVA should be used instead of pairwise comparisons with t-test.

Reply: We appreciate the reviewer's attention to proper statistical rigor and believe these updates improve the robustness of our analysis. In the revised manuscript, we have re-evaluated all statistical comparisons involving three or more groups. For these experiments, we have now applied one-way ANOVA to assess statistical significance. All relevant figure legends and methods have been updated accordingly. Pairwise comparisons using Student's t-test with Welch correction have been retained only for experiments with two-group comparisons, as originally indicated.

5) For all Western blot data, authors should provide information on the sample sizes (representatives of x experiments or only done once?)

Reply: We thank the reviewer for this important point. We have now updated all figure legends to clearly indicate that unless otherwise noted, all Western blot data shown are representative of at least three independent experiments. These revisions ensure clarity and reproducibility of our experimental data, and we appreciate the reviewer's suggestion to strengthen transparency.

6) The manuscript would greatly profit from thorough editing. There are a large number of typos, grammatical errors and long sentences that are difficult to understand.

Reply: We appreciate the reviewer's feedback regarding the clarity and readability of the manuscript. In response, we have carefully revised the text throughout the manuscript to correct grammatical errors, typos, and overly long or ambiguous sentences. The revised version has also been reviewed by co-authors with scientific editing experience to ensure clarity, conciseness, and consistency in tone and style. We thank the reviewer for highlighting this important issue and believe the revised manuscript is now significantly improved in readability and presentation.

Minor comments:

1) Introduction, page 3: "...activates downstream STAT proteins upon cytokine-stimulated receptor dimerization" should be corrected to "...activates downstream STAT proteins upon cytokine-stimulated conformational changes of receptor complexes".

Reply: We appreciate the reviewer's insightful comment. We agree with this correction and have revised the sentence in the Introduction (page 3) to read: "...activates downstream STAT proteins upon cytokine-stimulated

conformational changes of receptor complexes.”

2) Results, page 5: “.....Consistent with the human TYK2 results, knockdown of mouse TYK2 led to a high percentage of mammary organoids undergoing EMT with corresponding changes in EMT markers and invasion at low stiffnesses (Fig. 1h,i and Extended Data Fig. 1a)”. There are no data on EMT markers shown in the Figures. Please correct.

Reply: We thank the reviewer for pointing out this discrepancy. We agree that EMT marker data were not included in the figures. To correct this, we have revised the text on page 5 to read:“Consistent with the human TYK2 results, knockdown of mouse TYK2 led to a high percentage of mammary acini invade at low stiffnesses (Fig. 1h,i and New Extended Data Fig. 1b).”

3) Results, page 10: “.....involves interferon-induced heterodimerization between IFNAR1 (a direct binder of TYK2) and IFNAR2 (a direct binder of JAK1).....”. Correct to “.....involves heterodimerization between IFNAR1 (a direct binder of TYK2) and IFNAR2 (a direct binder of JAK1).....”.

Reply: We thank the reviewer for this correction. We have revised the text on page 10 (now page 11) accordingly to read:“...involves heterodimerization between IFNAR1 (a direct binder of TYK2) and IFNAR2 (a direct binder of JAK1)...”

4) Figure 3j: TYK2 levels strongly increase with increasing ECM stiffness in the sh-control, which is not commented by the authors and seems to contradict later results (e.g. Fig. 4a, Ext Data Figure 1a). Please clarify.

Reply: Harvesting 3D organoids from small coverslips for biochemical analyses is technically challenging due to carryover Matrigel in the extract. We have repeated this experiment several times and did not observe significant changes on TYK2 protein levels, as shown in **Extended Data Fig. 1e, Fig.2k, New Fig. 3i and New Fig. 3k, and Fig.4a.**

5) Figure 4f. Change figure caption to STAT3/5 inhibitor (instead of pan-STAT inhibitor).

Reply: We thank the reviewer for noting this inaccuracy. We have revised the caption of Extended Data Fig. 4f (now **new Extended Data Fig. 6f**) to read: “STAT3/5 inhibitor” (instead of “pan-STAT inhibitor”).

6) Results page 12/13: “Deucravacitinib treatment reduced phospho-STAT3 signal in primary tumors, indicating effective inhibition of the TYK2 kinase activity (Supplementary n. Fig. 6e).” Supplementary Figure 6e (i.e. Extended Data Figure 6e) shows something different. P-STAT3 is shown in Extended Data Figure 5d. Please correct.

Reply: We thank the reviewer for carefully noting this error. We agree that phospho-STAT3 data are presented in Extended Data Figure 5d (Now **New Extended Data Fig. 7b**).”, not Extended Data Figure 6e. We have corrected the text on pages 13 to read:“Deucravacitinib treatment reduced phospho-STAT3 signal in primary tumors, indicating effective inhibition of TYK2 kinase activity (**New Extended Data Fig. 7b**).”

7) Discussion, page 14: “At high stiffnesses of breast tumor tissues, TYK2 dissociates with IFNAR1 from the cell membrane,....”. It remained unclear whether IFNAR1 dissociates with TYK2 from the membrane or not.

Reply: We thank the reviewer for this important comment. To address this, we have revised the sentence on page 15 to read: “At high stiffnesses of breast tumor tissues, TYK2 dissociates from the cell membrane,” We have also expanded the Discussion (in reference to major comment 1) in Page 16 Paragraph 2.

Reviewer #3 (Remarks to the Author):

This study by Hu and Majeski et al. presents an extensive investigation into the role of TYK2 as a mechanosensitive suppressor of EMT and invasion in breast epithelial and tumor cells. The authors propose a noncanonical pathway in which soft ECM conditions maintain TYK2 at the membrane to repress TWIST1 nuclear localization via EPHA2 and LYN, independently of classical JAK–STAT signaling. The work includes stiffness-controlled 3D cultures, PDX-derived organoids, in vivo xenografts, and patient tissue analysis. The study is interesting, proposes a new mechanism by which tissue mechanics can regulate invasive behavior and metastasis, and is particularly relevant given the therapeutic use of TYK2 inhibitors to modulate immune responses in autoimmune disease. However, key mechanistic steps are dissected only in non-tumorigenic models and are not validated in PDX-derived organoids. Invasion quantification lacks cellular resolution, and the role of canonical mechanotransducers is not addressed. In addition, conclusions regarding the signaling axis and basal identity require stronger experimental support. Overall, this is a compelling study, but major revisions are required before publication in Nature Communications. Some of my detailed comments are summarized below.

Reply: We sincerely thank the reviewer for the thorough assessment of our work. We are pleased that the reviewer finds our study compelling and recognizes the significance of uncovering a noncanonical, mechanosensitive role of TYK2 in suppressing EMT and invasion, as well as its relevance to the clinical use of TYK2 inhibitors. We also appreciate the reviewer’s insightful concerns regarding the mechanistic validation in PDX-derived organoids, the resolution of invasion quantification, and the need to address canonical mechanotransducers and basal identity. Accordingly, we have performed additional analyses, including a) Incorporating validation of key mechanistic steps in PDX-derived organoids; b) Including new data and expanding discussion to address the role of canonical mechanotransducers. c) Clarify notions regarding the TYK2–IFNAR1–EPHA2–LYN–TWIST1 signaling axis and basal cell identity. We believe these additions and revisions have substantially strengthened the rigor and clarity of the manuscript.

Major Comments

1. Mechanistic validation in clinically relevant models is insufficient. The proposed TYK2-EPHA2-LYN-TWIST1 axis is dissected entirely in MCF10A and Eph4Ras models but generalized to patient-derived TNBC. In PDX-derived organoids, only invasion phenotypes are tested. In the in vivo analysis using PDXOs (e.g., Fig. 7j), the TWIST1 nuclear signal differences are not clear; quantitative analysis (nuclear/cytoplasmic ratios) and nuclear counterstaining are needed. Additionally, mechanistic validation in PDXOs, such as showing changes in EPHA2 phosphorylation and TYK2 localization, is essential to support the claim of a conserved metastasis suppressor mechanism in human tumors.

Reply: We thank the reviewer for the comment and fully agree that additional mechanistic validation in clinically relevant models is critical to strengthen our conclusions. To address this, we have included a number of new results below.

I). We have performed TWIST1 IHC staining with clearer nuclear counterstaining in the patient derived xenograft tumors (**New Fig.7j**). Quantitative analysis of TWIST1 nuclear staining is now included in **New Fig.7k**. These data confirm that TYK2 inhibition significantly increases TWIST1 nuclear localization in vivo. The TWIST1 antibody is only sensitive enough to detect endogenous TWIST1 protein by immunohistochemistry on tissue slides, not by multi-color immunofluorescence.

Response Fig. 9 (New Fig.7 j,k) : **a)** Immunohistochemistry staining for TWIST1 in patient-derived xenograft (PIM025) primary tumors treated with Vehicle control and TYK2 inhibitor deucravacitinib. Scale bar represents 50µm. **b)** Nuclear TWIST1 staining of vehicle and TYK2 inhibitor deucravacitinib treated Patient Derived Xenograft primary tumors (PIM025) were quantified and represented as percentages of total cells. Dots represent individual fields. ** p < 0.01.

II). To assess EPHA2 signaling in PDX-derived organoids, we orthotopically injected patient-derived xenograft (PIM025) tumor cells into the mammary fat pad of female NSG mice. After six weeks of tumor growth, tumors were harvested, and organoids were established. These PDX-derived organoids were then subjected to either

Response Fig. 10 (New Fig.3 k) : **a)** Bright field images of the control or TYK2-silenced PDX-derived organoids (PIM025) that were grown on 3D-PA gels and treated with vehicle control (DMSO) or the TYK2 inhibitor Zasocitinib(1µM) for 5 days. Scale bar represents 100µm. **b)** Lysates from the control or TYK2-silenced PIM025 PDX-derived organoids grown on 3D-PA gels and treated with treated with vehicle control (DMSO) or the TYK2 inhibitor Zasocitinib(1µM) for 5 days and immunoblot analysis of pEPHA2(897), total EPHA2, pSTAT1(Tyr701), total STAT1 and TYK2 expression in the lysates. GAPDH is used as a loading control.

TYK2 inhibition or shRNA-mediated TYK2 knockdown and cultured on 3D-PA gels. We evaluated EPHA2 S897 phosphorylation under various ECM rigidities and found that both TYK2 knockdown and pharmacological inhibition increased S897 phosphorylation of EPHA2 (**New Fig.3 k**). It's worth noting that tumor cells have elevated Erk kinase activity, which we found to be responsible for S897 phosphorylation (Fattet et al., 2020), so the basal S897 phosphorylation on EPHA2 is significantly higher in tumor cells than in MCF10A non-transformed cells. Consistent with what we reported in Fattet et al, 2020, phosphorylation of S897 on EphA2 alone is not sufficient to promote TWIST1 nuclear localization and EMT.

III). We show in human breast tumor tissue microarrays from our own analysis and from open access data on Human Protein Atlas that TYK2 is localized on the cell membrane in normal human breast tissues and delocalized to the cytoplasm in human breast cancer. We tested all commercially available TYK2 antibodies for immunofluorescent staining of endogenous TYK2 protein in PDX organoids in Matrigel 3D cultures and could not obtain reliable signal even though the best antibody we selected works on these PDX tumor sections. Working with 3D cultures in Matrigel, we have experienced several similar cases and think that the issue is that Matrigel absorbs certain primary antibodies due to non-specific binding, therefore it is currently not feasible to perform immunofluorescent staining of endogenous TYK2 protein in PDX organoids Matrigel culture.

2. Definition and quantification of invasion lack resolution. Invasion is inferred from morphological features (e.g., protrusions, spreading) in brightfield images, which is insufficient to distinguish true matrix invasion from acinar flattening or spreading. Higher magnification images or confocal z-stacks are needed. In spread conditions, for instance, in Figs. 1h, 1l, 2i, and S1e, overlapping acini/organoids complicate quantification. The authors should clarify how invasive fractions were determined, especially under these conditions.

Reply: We thank the reviewer for this valuable critique. We agree that relying solely on morphological features in brightfield images limits the resolution needed to distinguish true matrix invasion from acinar flattening or spreading. To further determine invasion, we performed immunostaining for Laminin V to examine basement membrane breaching. Consistent with what we reported in Wei et al., Nature Cell Biology, 2015, high stiffness promotes EMT and breach of basement membrane, so does blockade of TYK2 also promote EMT and invasion. We have added this data in the revised **Extended Data Figure 1b**.

Regarding the quantification of invasive fractions, invasive fractions were calculated as the percentage of total organoids displaying protrusive extensions into the matrix across at least three randomly selected, non-overlapping fields per well, with manual exclusion of ambiguous clusters or overlapping organoids. This information has now been described in more detail in the Methods section.

3. Canonical mechanotransducers are not addressed. The study does not investigate integrin-mediated mechanotransduction, including integrin expression, FAK/SRC activation, or YAP localization, which are key regulators of stiffness-driven invasion. Given the potential convergence between EPHA2, TYK2, LYN, and integrins, this pathway should be considered—for example, by assessing focal adhesion signaling or YAP localization following TYK2 inhibition. Moreover, the role of the ECM warrants deeper discussion. Although

TYK2 delocalization at high stiffness is described, the upstream mechanical inputs regulating this process remain undefined. Future work should explore whether force-sensitive mechanisms—such as integrin signaling, cytoskeletal tension, or membrane compartmentalization—govern TYK2 localization and activity, particularly in physiologically relevant, collagen-rich 3D environments. This represents a significant gap in understanding how TYK2 regulates invasion in a mechanically dynamic context.

Reply: In response to the reviewer’s suggestions, we examined the most well-known mechanotransducers, including integrins, FAK/SRC activation and YAP localization in regulating TWIST1 mechanotransduction. In our two previous publications, Wei et al., *Nature Cell Biology* 2015 and Fattet et al., *Developmental Cell* 2020, we reported that YAP translocates from the cytoplasm to the nucleus at high ECM rigidities due to the weakening of cell-junctions and changes in actin cytoskeleton as the consequence of EMT, However, YAP does not control TWIST1 subcellular localization in response to stiffness changes.

We next examined focal adhesion signaling via FAK phosphorylation in human MCF10A acini upon TYK2 knockdown. When epithelial cells undergo EMT, these cells lose hemidesmosomes-mediated basement membrane attachment and switch to focal adhesions. Consistent with this notion, we observed increased pFAK signal at 5700Pa compared to at 320 Pa. Upon TYK2 knockdown, cells undergo EMT and presented increased pFAK at 320Pa, similar to control cells at 5700Pa. Inhibition of LYN to block TWIST1 nuclear localization and EMT induced by TYK2 knockdown also reduced pFAK, suggesting that FAK activation is a consequence of EMT induction at high ECM stiffnesses.

To further address this reviewer's comment and understand whether FAK regulates TYK2 kinase activity in response to ECM rigidity change, we treated MCF10A acini with a FAK inhibitor VS-4718. Together with what we reported in Fattet et al, *Developmental Cell* 2020, VS-4718 potently inhibited pFAK signal, but did not block TWIST1 nuclear localization nor EMT at 5700Pa. Importantly, FAK inhibition did not block reduction of TYK2 kinase activity indicated by pTYK2 Tyr1054/1055 at high ECM rigidities (Revised in **New Extended Data Figure 5c-f**). Altogether, these results show that high stiffness activates FAK as a consequence of EMT, but FAK does not regulate TYK2 in response to ECM stiffness.

In terms of the involvement of Src in the TWIST1 mechanotransduction, In Fattet et al., *Developmental Cell*, 2020, Fig.2m-o and Fig. S4A, we carefully compared SRC vs. LYN on TWIST1 phosphorylation and found that LYN, but not SRC is responsible for TWIST1 phosphorylation and activation at high ECM rigidities. Furthermore, Fig. S4 shows that Src kinase inhibition by PP2 did not block TWIST1 nuclear localization nor cell invasion at high ECM rigidities. Therefore, we concluded that LYN, but not SRC, is uniquely activated to phosphorylate TWIST1 at high ECM rigidities. We included these published data (**Fattet et al, *Developmental Cell* 2020**) here for this reviewer's information.

[FIGURE REDACTED]

Response Fig. 14 (Cite figure in Fattet et al, *Developmental Cell* 2020): Cite Figure 2. M) Lysates from 3D-cultured MCF10A cells were subjected to LYN or SRC immunoprecipitation and immunoblotted as indicated. **N)** Relative Protein levels in M were quantified by densitometry (presented as the ratio of phospho-Y416 to total LYN/SRC levels). **O)** LYN or SRC immunoprecipitates from 3D-cultured MCF10A cells were used for cold in vitro kinase assay. Reactions were immunoblotted as indicated. **Cite Extended Data Figure 4.A)** Eph4Ras cells were grown on 3D-PA gels with different rigidities for 3 days in the presence of 1 μ M Bafetinib, 500 nM FAK inhibitor VS-4718, 1 μ M SRC inhibitor PP2 or vehicle control, fixed and stained for Twist1 (green) and DAPI (blue). Scale bar = 25 μ m.

To further understand the role of various integrins in the TWIST1 mechanotransduction as suggested by both Reviewer 1 and 3, we have performed a number of integrin blockade experiments (please see responses to Reviewer 1 Point #1). In summary, only the anti-integrin β 1 blocking antibody AIIB2 that blocks cell attachment to fibronectin, collagen-I, collagen-IV, and laminin (Hall et al, *JCB*, 1990) could potently suppress invasion in all rigidities in the control cells and in the cells with TYK2 knockdown. However, inhibition of neither FAK nor SRC did so. Consistent with this data, we found that the anti-integrin β 1 blocking antibody AIIB2 treatment blocked TYK2 inactivation (**New Extended Data Figure 5c,d,e,f**) and TYK2 membrane mis-localization at high ECM rigidities (**New Extended Data Figure 5g,h**). Following this reviewer's suggestion, we have included a **new extended Figure 5** to report all these results to further strengthen our study.

Response Fig. 15 (New Extended Data Figure 5g,h): a) MCF10A acini overexpressing FLAG-TYK2 WT and treated with β1 integrin AIIB2 blocking antibody or vehicle control that were grown on 3D-PA gels. and then immunostained for TYK2 (red) and DAPI (blue). Scale bar represents 25μm. **b)** Plots of DAPI and TYK2 intensity profiles plotted as a function of distance across tissue sections versus their pixel intensities in human MCF10A acini overexpressing FLAG-tagged TYK2 and treated with β1 integrin AIIB2 blocking antibody or vehicle control that were grown on 3D-PA gels.

4. ECM composition does not model physiological invasion contexts. The stiffness-controlled ECM model lacks fibrillar collagen I, a key component of breast tumor stroma typically invaded by cancer cells. Coated PA gels with non-fibrillar collagen I do not recapitulate 3D ECM architecture. The role of TYK2 in fibrillar collagen-rich matrices should be tested, and these limitations clearly acknowledged. The authors may also consider testing the effect of TYK2 knockdown on invasion using primary tumors isolated from their in vivo models.

Reply: We agree with the reviewer that using fibrillar collagen I to study tumor cell migration and invasion after tumor cells have already undergone EMT and breached through basement membrane is critical to understand cell migration on physiologically relevant ECM. The current study focuses on understanding how ECM rigidity activates the EMT program to enable epithelial tumor cells breach basement membrane before these tumor cells are in contact with fibrillar collagen I that is not present in basement membrane. Therefore, we used the Matrigel overlay culture system, which is widely used to study epithelial cells undergoing EMT and well-described by Debnath and Brugge, 2005, Nature Reviews Cancer (Modeling Glandular Epithelial Cancers in Three-Dimensional Cultures). In this Matrigel culture system, mammary epithelial cells form 3D acini with intact basement membrane. The data from our study show that TYK2 functions to block EMT and basement membrane breaching by maintaining TWIST1 cytoplasmic localization at low ECM stiffness. Whether inactivation of TYK2 has additional roles in regulating tumor cell migration and invasion in fibrillar collagen I after EMT activation and basement membrane breaching is an interesting, but different question from the focus of this study on its role in EMT.

We also appreciate the suggestion to examine TYK2 knockdown in primary tumors isolated from in vivo models. Indeed, all TNBC PDX-derived organoids studies presented in this study are organoids freshly

isolated from the TNBC PDX tumors and never being passaged in vitro. We thank the reviewer again for this constructive feedback and have revised the manuscript to clarify the scope of our current model.

5. JAK-STAT signaling independence is not conclusively shown. The authors show that JAK/STAT inhibition does not phenocopy TYK2 loss but do not assess whether TYK2 knockdown alters JAK1/2 or STAT3/5 phosphorylation. Compensatory effects could influence EMT independently of canonical transcriptional activity. This should be evaluated under varying stiffness conditions.

Reply: We thank the reviewer for raising this interesting point. Our data show that pharmacological inhibition of JAK1/2 or STAT3/5 does not induce EMT at low ECM rigidities. Following the reviewer's suggestion, to determine whether TYK2 knockdown could lead to compensatory activation of JAK1/2, which subsequently activates EMT, we treated TYK2-knockdown mammary acini with the JAK1/2 inhibitor AZD1480 and found that inhibition of JAK1/2 also did not block EMT induced by TYK2 loss at all ECM stiffnesses tested (**New Extended Data Figure 2e, f**).

6. Experimental rationale is unclear in several cases. For example: Why is immunoprecipitation used to detect phospho-TYK2 instead of standard western blot? What is the rationale for using TYK2 overexpression and PLA assays?

Reply: We thank the reviewer for pointing out the need to clarify the experimental rationale and have revised the manuscript accordingly.

1. Use of immunoprecipitation to detect phospho-TYK2: This approach is commonly used for phospho-specific antibodies due to their low sensitivity and potential cross-reactivity to Tyr phosphorylation sites on unrelated proteins with similar molecular weight in whole cell extracts. We used immunoprecipitation (IP) followed by immunoblotting to specifically enrich endogenous TYK2 protein and then detect phosphorylated TYK2. Similarly, the antibody recognizing Y416 phosphorylation on LYN/Src recognizes the conserved Tyr phosphorylation site on all Src-family kinases, including LYN. Therefore, we first immunoprecipitated LYN and then probed with phospho-Y416 antibody to specifically detect this phosphorylation on LYN, but not other Src family kinases. We performed these comprehensive analyses to ensure utmost specificity and rigor.
2. Using Tyk2 overexpression: We tested all TYK2 antibodies commercially available and none of them works efficiently for co-immunoprecipitation and immunostaining of endogenous TYK2 in the Matrigel 3D culture. Therefore, tagged TYK2 is expressed to allow such analyses.
3. PLA assay is a sensitive technique to detect endogenous protein interactions in situ, so the PLA signal could reveal not only the physical interaction between two proteins, but also the subcellular localization of this interaction, which is not feasible by conventional co-IP. In our study, we used the PLA assay to detect specific association between endogenous TWIST1 and G3BP2 proteins in the cytoplasm at low ECM stiffnesses.

7. Incomplete characterization of invasion in vivo. In Fig. 6i, red arrows indicate putative invasive cells, but these may represent sectioning artifacts. This panel is not described in the Results section. More broadly, it is unclear how invasion and dissemination are distinguished from local displacement.

Reply: We thank the reviewer for raising this point. Local invasion is defined as individual lesions that are completely separated from the primary tumor mass in the mammary gland. The images are from freshly dissected mouse mammary glands that are directly imaged under dissection microscope without sectioning. As marked by scale bars, these lesions are ~mm away from the primary tumor mass.

8. Effects on proliferation not assessed. In Fig. 1i, TYK2 knockdown organoids appear larger, suggesting altered growth. This is not tested in PDXOs. Similarly, for in vivo tumors (Fig. 6b, f), tumor weight alone is insufficient to assess growth effects. Additional metrics such as tumor volume and proliferation markers (e.g., Ki-67) should be included. Time-course growth analyses in vitro are also recommended. Tumor volume could be included in the supplemental data.

Reply: We thank the reviewer for raising this important point. To better assess the effects of TYK2 knockdown on proliferation and tumor growth, we have included additional analyses in the revised manuscript. For the in vivo tumor studies (**new Fig. 6 and new Fig. 7**), we now provide tumor volume measurements over time in addition to endpoint tumor weight. These longitudinal measurements allow a more dynamic evaluation of tumor growth kinetics and provide a clearer assessment of potential proliferation differences. According to the tumor growth kinetics in MCF10 DCIS and PIM025_PDX xenograft tumor, TYK2 inhibition have no significant influence the tumor growth, for MCF10 DCIS xenograft tumors, the P value is 0.4531, PIM025 patient-derived xenograft tumors, the P value is 0.2908. Furthermore, we performed Ki-67 immunostaining to evaluate proliferation at the cellular level. Quantification of Ki-67-positive cells revealed that TYK2 knockdown did not significantly alter the percentage of proliferating tumor cells. Together, these data demonstrate that while TYK2 knockdown influences tumor invasion and metastatic potential, it does not significantly impact tumor cell proliferation.

Response Fig. 17 (New Extended Data Figure 1c,d; New Extended Data Figure 7a,c-e; New Extended Data Figure 8f-h):
Panel a and b correlate with New Extended Data Figure 1c,d. **a)** Control or TYK2-silenced human MCF10A acini were grown on 3D-PA gels for 5 days and then immunostained for Ki67 (red) and DAPI (blue). Scale bar represents 25mm. **b)** Ki67 staining of Control or TYK2-silenced human MCF10A acini were quantified and represented as the percentage of total cells. Dots represent individual fields. P-value:0.037, ns, not significant. **Panel c-f correlate with New Extended Data Figure 7a,c-e.** **c)** Tumor growth curves of MCF10DCIS xenografts expressing control or TYK2 shRNA(s) (n = 10 tumor per group), no difference between groups (P = 0.5715). **d)** Tumor growth curves for MCF10DCIS xenograft primary tumors from vehicle and the TYK2 inhibitor deucravacitinib treated groups (n = 13 tumor per group), no difference between groups (p = 0.4531). **e)** Immunohistochemistry staining analysis of Ki67 in MCF10DCIS xenograft primary tumors from vehicle and TYK2 inhibitor deucravacitinib(5 weeks) treated groups. Scale bar represents 50µm. **f)** Ki67 staining of MCF10DCIS xenograft primary tumors from vehicle and the TYK2 inhibitor deucravacitinib (5 weeks) treated groups were quantified and represented as percentages of total cells. Dots represent individual fields. ns, not significant. **Panel g-i correlate with New Extended Data Figure 8f-h.** **g)** Tumor growth curve for Patient Derived Xenograft primary tumors (PIM025) from vehicle or the TYK2 inhibitor deucravacitinib treated groups (n = 16 tumor per group), no difference between groups (p = 0.2908). **h)** Immunohistochemistry staining analysis of Ki67 in Patient Derived Xenograft primary tumors (PIM025) from vehicle and TYK2 inhibitor deucravacitinib treated groups. Scale bar represents 50µm. **i)** Ki67 staining of Patient Derived Xenograft primary tumors (PIM025) from vehicle or the TYK2 inhibitor deucravacitinib treated groups were quantified and represented as percentages of total cells. Dots represent individual fields. ns, not significant.

9. Terminology and labeling inaccuracies. MCF10A-based acini are referred to as “human basal mammary organoids.” This is misleading; these are not true organoids. Text and figure labels should be revised accordingly.

Reply: We thank the reviewer for pointing out this terminology issue. We changed the name to “acini” to replace “organoids” as the reviewer suggested. As discussed above, the basal B subtype of breast cells already express elevated EMT factors due to pre-existing unknown genetic/epigenetic changes. The expression of these factors in MCF10A cells has been previously documented in a comprehensive analysis of EMT factor expression in 51 breast cancer cell lines (Blick et al, 2008, Clin. Exp. Metastasis). MCF10A has been classified as basal subtype of breast epithelial cells based on RNA-seq data.

10. TYK2 overexpression vs. endogenous expression not clearly distinguished. In localization experiments, the authors must clarify whether TYK2 is overexpressed or endogenous. Unless stated otherwise, such experiments should rely on endogenous TYK2.

Reply: We thank the reviewer for this comment. We have carefully clarified the figure legends and text to specify endogenous TYK2 detection or TYK2 overexpression constructs. If appropriate antibodies for immunoprecipitation and immunostaining are available, we always use them to detect endogenous proteins in all our studies. As discussed in point #6, we have tried all available TYK2 antibodies, but none of them is efficient for co-IP experiments (the pull-down efficiency is below 5%) and Immunofluorescent staining in 3D Matrigel cultures (some antibodies bind to Matrigel non-specifically). So, we used tagged TYK2 for these specific studies. These instances are now clearly labeled in the revised figure legends and Results text.

11. Mechanistic assertions lack critical controls.

i) The EPHA2–S897 mechanism would be strengthened by using a phosphomimetic EPHA2-S897E mutant to test sufficiency for EMT and invasion.

Reply: We appreciate the reviewer’s insightful question. we tested both the EPHA2-S897A and EPHA2-S897E mutants. The S897A mutant failed to rescue LYN activation and EMT, as reported in Fattet et al, Developmental Cell, 2020, while the S897E mutant was also not sufficient to induce EMT or invasion (unpublished data). Based on these results, we concluded that S897 phosphorylation is necessary for stiffness-induced EMT. Since the glutamate mutation sometimes does not fully mimic the phosphate group on a tyrosine, we currently have an independent project to further study EPHA2 in mechanotransduction, which will provide important insights in the future.

ii). Co-localization of TYK2, IFNAR1 and EPHA2 (and phospho-EPHA2) is not shown; this is required to support membrane-dependent regulation as the authors claim that: “We show that at low matrix stiffness of normal mammary glands, TYK2 localizes on the cell membrane by binding to IFNAR1 and associates with EPHA2, thus blocking EPHA2 S897 phosphorylation. At high stiffnesses of breast tumor tissues, TYK2 dissociates with IFNAR1 from the cell membrane, thus allowing EPHA2 S897 phosphorylation and downstream activation of the TWIST1-driven EMT and invasion.”

Reply: IFNAR1 and EPHA2 are both transmembrane receptors. **Fig. 5j,k** shows that TYK2 protein localizes on the cell membrane via IFNAR1 association at low stiffness. Since staining can only show whether these proteins are present on the cell membrane, but not whether they form a complex, we have now performed additional co-IP experiments: TYK2 was immunoprecipitated from acini cultured on soft versus stiff matrices and then probed for both EPHA2 and IFNAR1. We found that indeed all three proteins are co-immunoprecipitated at low stiffness, and the complex dissociates at high stiffness. Importantly, we found that knockdown of IFNAR1 led to the dissociation between TYK2 and EPHA2, further supporting our conclusion. These results are presented below and in **New Fig. 5l,m**.

Following the reviewer’s suggestion, we also revised the statement as the following “We show that at low matrix stiffness of normal mammary glands, TYK2 binds to IFNAR1 on the cell membrane and associates with EPHA2, thus blocking EPHA2 S897 phosphorylation. At high stiffnesses of breast tumor tissues, TYK2

dissociates from the cell membrane, thus allowing EPHA2 S897 phosphorylation and downstream activation of the TWIST1-driven EMT and invasion.

iii) In several westens (e.g., Fig. 3j–k), total EPHA2 appears increased at high stiffness or upon TYK2 inhibition. These changes should be quantified and independently replicated to distinguish phosphorylation from total protein level changes. Similarly, TYK2 signal intensity appears reduced at high stiffness (e.g., Fig. S3b, Fig. 4b); the authors must distinguish between protein re-localization and downregulation.

Reply: We thank the reviewer for this comment. We agree that total EPHA2 protein appears to increase a bit upon increase in EMC stiffness. EPHA2 has also been reported to be a direct transcriptional target of the MAPK/ERK pathway by the Frank McCormick group (Macrae et al., Cancer Cell 2005). High ECM stiffness promotes Erk activation to not only drive EPHA2 S897 phosphorylation, but also increase EPHA2 protein levels (Fattet et al., Developmental Cell, 2020). For all phosphorylation signals, we calculate the phosphorylation signal over total protein signal to distinguish phosphorylation from total protein level changes.

On immunostaining images, diffuse cytosolic staining often appears weaker than sharp membrane-localized signal. This reflects differences in local protein concentration rather than differences in total protein abundance because the cytoplasm volume is much larger than the plasma membrane in a cell. We have carefully determined TYK2 protein levels by western blotting analyses and did not observe significant changes on TYK2 protein levels, as shown in **Extended Fig. 1e, Fig.2k, New Fig. 3i, New Fig. 3k, and Fig.4a**. As shown in **Extended Data Fig. 4a**, TYK2 mRNA levels also do not exhibit significant changes across different rigidities. Therefore, we conclude that TYK2 protein and mRNA levels do not change at different rigidities.

12. Some claims are not supported by the data shown. For example, on page 5 the authors state: “Knockdown of TYK2 potently induced EMT, as shown by increased Vimentin and decreased E-cadherin.” However, these markers are not shown. The cited figures quantify invasive acini rather than protein expression. If this conclusion is based on previous literature, this should be explicitly stated.

Reply: We thank the reviewer for pointing out this issue. We apologize for the confusion caused by placing the figure citation at the end of the sentence. The expression of EMT markers (increased vimentin and decreased E-cadherin) is indeed shown by both immunofluorescence and western blot analyses in Fig. 1f and Fig. 1g. We have moved the figure citation directly next to this phrase in the revised manuscript.

13. TYK2 function not linked to basal cell identity. The term “basal-subtype organoids” is used without showing basal marker expression. No data are provided on KRT14, KRT5, or p63 in MCF10A or PDXOs. The authors should validate epithelial cell identity and assess whether TYK2 function correlates with basal marker

expression. Is TYK2 differentially expressed or localized in invasive cells compared to cells within the acinar body?

Reply: We thank the reviewer for this comment. We have now performed immunofluorescence staining for KRT5 in TNBC PDOs to demonstrate their positivity (**New Extended Data Figure 2i**). Also the MCF10A acini models have been classified to be basal subtype based on their gene expression profiling by an independent group (Neve et al., Cancer Cell, 2006). It is important to note that TYK2 is highly expressed in luminal mammary epithelial cells and localized at the cell membrane in normal luminal epithelial cells, as shown in Fig. 4 and Extended Data Fig. 4. For luminal epithelial cells to undergo EMT, stroma factors, such as TGF-beta and hypoxia are needed to first induce the expression of EMT transcription factors, such as TWIST1. But at low ECM stiffness, TWIST1 protein is restricted in the cytoplasm due to TYK2 mechanotransduction. Increasing ECM stiffness inactivates TYK2 and leads to nuclear localization of TWIST1, which leads to EMT. Therefore, loss of TYK2 membrane localization in normal luminal epithelial cells alone is not sufficient to induce EMT and convert cells into a basal subtype without induction of EMT transcription factor gene expression by additional signals such as TGF-beta. This rationale explains why we employed basal-subtype mammary cells and PDX models in our study, as these already express endogenous EMT transcription factors, unlike normal luminal epithelial cells.

Minor Comments

1. Clarify drug concentrations used in Fig. 1b–c and their relevance.

Reply: We have added the exact concentrations and treatment durations of the JAK/STAT inhibitors used in Fig. 1b–c to both the figure legend and Methods section, and provided justification based on previously published IC50 values and dose–response pilot studies.

2. Improve figure referencing and ordering; panel arrangement (horizontal vs. vertical) and citation sequence (e.g., S2b before S1a) are inconsistent.

Reply: We thank the reviewer for highlighting the inconsistencies in figure ordering and panel arrangement. We have carefully revised the supplementary figures and reorder supplementary figures to ensure that: a) Panels are arranged consistently (horizontal or vertical, depending on clarity of data presentation). b) Figure panels are cited in sequential order. c) Cross-references between the main and supplementary figures are now consistent and aligned with journal formatting guidelines. These revisions improve readability and maintain logical flow between figure presentation and textual descriptions.

3. Quantification of PIM046 invasion (Fig. S1f) is not referenced in the text.

Reply: We thank the reviewer for noting this omission. We have now cited the quantification of PIM046 invasion (**new Extended Data Fig. 2d**) in the Results section to ensure the data are properly referenced. The revised text reads: “Consistently, TYK2 inhibition also promoted invasion of the PIM046 patient-derived xenograft organoids (Extended Data Fig. 2d).”

4. Emphasize and clarify the differences between MCF10A (non-tumorigenic) and tumor-derived models in the main text. Related to that, consider including a statement in the Discussion addressing the potential risk that TYK2 inhibition might promote invasive behavior, and possibly metastasis, in otherwise non-transformed epithelial tissues.

Reply: We thank the reviewer for this suggestion. We have revised the Results and Discussion sections to more clearly distinguish findings from the non-tumorigenic MCF10A model versus the tumor-derived models. Specifically, we now emphasize in the text that MCF10A acini represent a non-transformed epithelial model, whereas the PDX and TNBC organoid models capture tumor-derived invasive behavior. For non-transformed cells, additional genetic alterations are needed for them to gain tumorigenic ability to form metastases. We have added a statement in the Discussion acknowledging that TYK2 inhibition could promote invasive behavior in non-transformed basal-subtype epithelial cells.

5. Show knockdown efficiency for IFNAR1, IFNAR2, and TYK2 in all relevant models.

Reply: We have provided knockdown efficiency data for IFNAR1, IFNAR2, and TYK2 in all relevant models in **Fig. 1g, k, extended Fig. 1b, Fig. 3e,3i, 3k, Fig. 5l, 5m, Extended Fig. 6i, 6j, 6m, 6n, 6q, Fig. 6a, 6m, and Fig. 7l**. We also assembled the most critical knockdown validation data in Response Fig. 21 below.

6. Validate the efficacy of IFNAR1-blocking antibodies.

Reply: We thank the reviewer for this suggestion. As shown in the **new Extended Data Figure 6d**, IFN- α/β treatment activated STAT1 signaling, which was significantly blocked by IFNAR1-blocking antibodies. These results confirm the efficacy of the IFNAR1-blocking antibodies used in our study.

Response Fig. 22 (New Extended Data Figure 6d): Immunoblot analysis of pSTAT1 and STAT1 were performed on lysates from MCF10A cells unstimulated or stimulated with IFN α/β and treated with IFNAR1 blocking antibody as indicated. GAPDH is used as a loading control.

7. Some immunostainings are difficult to interpret (e.g., E-cadherin in Fig. 1f). Fibronectin appears increased in invasive conditions, but comparing monolayers to acini may confound interpretation due to differences in antibody penetration. The authors should include single-cell controls or validate penetration, as done in Fig. S3d.

Reply: We thank the reviewer for this comment. As shown in Fig. 1f, E-cadherin is highly expressed and localized at cell–cell junctions in acini cultured at low stiffness, whereas its level is reduced in invasive sheets at high stiffness. These results demonstrate that our permeabilization protocol effectively allows the antibodies to penetrate the acini and produces strong junctional E-cadherin signals. E-cadherin is the key protein at adherens junctions to mediate interaction between two cells and gets endocytosed for degradation when adherens junctions are compromised. Therefore, E-cadherin is not examined in a single cell that lacks cell-cell junctions. Fibronectin is an extracellular matrix protein present in basement membrane in epithelial acini, so there is no intracellular signal inside the acini. To further validate the IF results, **Fig. 1g** provides western blotting data demonstrating reduction of E-cadherin and increase of vimentin protein level.

8. Include time-lapse imaging or t = 0 frames for invasion assays, where feasible.

Reply: We thank the reviewer for this suggestion. Activation of EMT requires TWIST1-induced changes in gene expression and then downstream morphological changes. This process requires live 3D imaging for at least 5 days to observe EMT and invasion. Such experiments necessitate an exceptionally stable and sophisticated long-term imaging system, which is not currently accessible to us.

9. Some mechanistic conclusions rely on cited literature rather than directly shown data.

Reply: We thank the reviewer for this helpful comment. We have carefully revised the text to clearly distinguish our experimental findings from mechanistic conclusions that are inferred from previously published studies. Where appropriate, we have now qualified such statements and cited the original sources to ensure accurate attribution.

10. Include methodology for PDX organoid culture and maintenance.

Reply: We thank the reviewer for this comment. We have added detailed methodology for PDXO derivation, culture, and ECM embedding in the Methods section. This includes descriptions of media composition, matrix stiffness conditions and maintenance to ensure reproducibility.

11. Is the observed effect on TYK2 localization and activity upon increasing stiffness specific to PA gels coated with collagen I?

Reply: We thank the reviewer for this important question. To address whether the observed effect is specific to PA gels, we also collaborated with bioengineers to establish other hydrogel systems that mimic breast tissue rigidities, such as methacrylated HA gels, and observed similar stiffness-induced EMT (Ondeck et al., PNAS 2019). Importantly, we also observe changes in TWIST1 and TYK2 localization in human breast tumor tissues, supporting that this effect is not limited to PA gels. In our system, PA gels are functionalized with collagen I on the surface, while the mammary acini are embedded in Matrigel. **New Extended Data Figure 1b** and in Fattet et al., 2020 (Fig. 1H), the acini are surrounded by a layer of basement membrane secreted by human organoids. This is validated by Laminin V staining, which specifically recognizes human laminin but not the mouse laminin

presents in Matrigel. Furthermore, consistent changes in TYK2 localization were also observed in human breast cancer samples (**Fig. 4** and **Extended Data Fig. 4**), further supporting that this phenomenon indeed occurs in human breast tumor samples.

We thank all the reviewers for their positive comments about this study and are grateful for all your insightful suggestions that have helped us to substantially strengthen this manuscript.

Reviewer #1 (Remarks to the Author):

The authors have addressed my previous concerns.

Reviewer #2 (Remarks to the Author):

The authors have thoroughly and sufficiently addressed all my concerns and comments.

Reviewer #3 (Remarks to the Author):

The authors have adequately addressed most of my previous comments. This is a very nice and well-executed manuscript. A few minor points remain:

1. The explanation that TYK2 immunofluorescence is not feasible due to antibody absorption by Matrigel is not fully convincing. Organoids can be efficiently and rapidly (few seconds) recovered from Matrigel using cold media, followed by fixation and staining in suspension or embedding. This approach is routinely used for organoid immunostaining and would allow visualization of TYK2 localization without matrix interference.

Response: We appreciate this helpful suggestion. Indeed, our protocol always performs 30min cold wash to remove bulk Matrigel while maintaining cell attachment before fixation and staining, which is critical for generating clean IF signals in Matrigel 3D cultures. However, even complete suspension recovery could not remove all the Matrigel that attaches to the basement membrane of individual suspended acini. More importantly, at high stiffnesses, cells undergo EMT, lose basement membrane and acini structures and invade as single cells and sheets. The suspension method could not recover such invading structures for IF staining. Throughout the manuscript, we show that the overexpressed TYK2 responds to the mechanical signals and various pharmacological and genetic perturbations similarly to endogenous TYK2. The TYK2 immunostaining data in human breast tissues generated by us and by Human Protein Atlas independently all support mislocalization of endogenous TYK2 protein in invasive breast cancer. In the text, we ensure to clearly state “in human MCF10A acini expressing FLAG-tagged human TYK2, TYK2 showed robust plasma membrane localization at low stiffness, but largely disappeared from the cell membrane at high stiffness (Fig. 4b,c)”.

2. The statement that YAP does not control TWIST1 subcellular localization seems too narrow and insufficiently supported by data. YAP and TWIST1 are known to regulate each other's expression and activity in multiple systems, including breast cancer. Even if YAP does not directly influence TWIST1 localization, it could modulate TWIST1 levels through transcriptional and mechanotransduction pathways; which could affect TWIST1 distribution. The authors should clarify whether YAP activity affects TWIST1 localization or expression in their system, or discuss YAP's potential role in the Discussion. This point is particularly relevant given that the pathway is dependent on β 1-integrin.

Response: We appreciate the reviewer's insightful comment and agree that YAP and TWIST1 could potentially interact functionally in response to changes in ECM stiffness. Pharmacological or genetic inhibition of YAP/TAZ potently suppress cell proliferation and survival in various TNBC cells, which complicates the study of EMT and cell migration. As this manuscript focuses on the role of TYK2 as a

breast cancer metastasis suppressor, we plan to collaborate with a YAP expert group to perform independent studies to examine all possible links between YAP and TWIST1 thoroughly. Align with the scope of this manuscript focusing on TYK2, we added a new discussion of YAP in the context of FAK inhibition on TYK2 and TWIST1. While inhibition of FAK is reported to suppress YAP activation (Feng et al., Cancer Cell, 2019), inhibition of FAK did not affect TYK2 activation nor TWIST1 subcellular localization at various ECM rigidities. These data suggest that ECM rigidities impinge on YAP vs TWIST1 via distinct signaling regulators and inhibition of YAP/TAZ via FAK inhibition is not sufficient to impact the TYK2/TWIST1 mechanotransduction. We thank the reviewer for this suggestion and have added an entire paragraph to discuss this point (Page 15-16), as below in response to point #3.

3. The new integrin interference data clearly indicate that β 1-integrin engagement is required for stiffness-dependent TYK2 inactivation and EMT induction, whereas FAK activity is dispensable for this step. This mechanistic conclusion should be explicitly stated in the manuscript to ensure alignment between the data and interpretation.

Response: We thank the reviewer for highlighting this important conclusion. To further highlight this, we put it as a new section in the Results and added a section title “ β 1-integrin engagement, but not FAK is required for high stiffness-dependent TYK2 inactivation and EMT induction”. We have also added a new paragraph in the Discussion sections accordingly: “We found that while β 1-integrin engagement is essential to control the TYK2 kinase activity in response to different ECM rigidities, the integrin downstream kinase FAK is dispensable for TYK2 mechanotransduction. This result is consistent with what we reported previously that β 1-integrin, but not FAK is required for high stiffness-induced TWIST1 nuclear localization and EMT^{9,10}. The YAP/TAZ transcription activators are also known to be regulated by ECM rigidities⁴⁸. Interestingly, not only β 1-integrin is required for YAP activation⁴⁹, but the FAK kinase has been shown to activate YAP nuclear localization via phosphorylation of MOB1 to inhibit core upstream Hippo signaling⁵⁰. These distinctions demonstrate that FAK inhibition, which inhibits YAP/TAZ, does not impact TYK2/TWIST1 mechanotransduction and underscore that ECM rigidities impinge on YAP vs TWIST1 via distinct signaling regulators. Future work is needed to reveal how β 1-integrin directly or indirectly impinges on TYK2/TWIST1 mechanotransduction in a FAK-independent manner in response to ECM rigidities.”

4. The authors’ response clarifies that the study focuses on how ECM rigidity enables basement membrane breaching during EMT. However, this is not clearly articulated in the current manuscript. The authors are invited to explicitly define basement membrane breaching as the central biological process under investigation to strengthen conceptual focus and clarify the relevance of the Matrigel system.

Response: We thank the reviewer for bringing this remaining misunderstanding to our attention. The central biological process under investigation in this manuscript is induction of EMT by high EMC stiffness. All normal epithelial cells, including carcinoma in situ, are surrounded by basement membrane, not type I-collagen matrices. EMT induction described in this study is triggered by high stiffness-activated nuclear localization of the EMT-inducing transcription factor TWSIT1, which in turns induces the EMT cellular and morphological changes, including loosening cell-cell junctions and breaching the underlying basement membrane. Therefore, we examine both E-cadherin-mediated adherens junctions and beach of basement membrane as cellular readouts of EMT. We have revised the Introduction, Results and Discussion to emphasize that our work specifically addresses how ECM rigidity controls EMT initiation. The text now

clearly explains that the 3D Matrigel system serves as a physiologically relevant model that mimics the biochemical and mechanical ECM environment of the mammary epithelium.

Introduction(Page 3-4): “Human and mouse basal-like mammary epithelial cells, which are known to express a gene signature enriched with the Epithelial-Mesenchymal Transition (EMT) program, undergo EMT and invade surrounding matrices when cultured at tissue stiffnesses between 1000-5700 Pascals (Pa), which were detected in some human breast tumors^{8-10,23}. In contrast, these same cells form mammary acini with adherens junctions and intact basement membrane under approximately 150-320Pa observed in normal breast tissue even though they express endogenous EMT transcription factors^{9,10}”.

Main Text (Page4): “In a search for novel molecular regulators linking ECM mechanical cues to EMT and invasion, we utilized a hydrogel Matrigel overlay 3D mammary acini culture system as the following: 1) the base layer is the polyacrylamide(PA) hydrogel with calibrated elastic moduli ranging from 150-320Pa present in normal human breast tissues to 1100-5700Pa observed in some stiff breast tumors; 2) the PA hydrogel is crosslinked with collagen I to enable mechanical coupling across the interface; 3) mammary acini embedded in Matrigel, which provides basement membrane-like matrices, are seeded on top of the hydrogel to be exposed to stiffness cues coming from the underlying collagen I-coated PA gel. The 3D model employed here recapitulates the biochemical and mechanical ECM microenvironment of breast carcinoma in situ, enabling mechanistic dissection of rigidity-dependent signaling events that initiate EMT. At low ECM rigidities, human basal-like non-tumorigenic MCF10A cells form mammary acini surrounded by basement membrane, while they undergo EMT and invade at high rigidities (Fig. 1a,f and Extended Data Fig. 1b).”

5. The architecture and rationale of the experimental setup should be described more clearly. From the response, acini are embedded in Matrigel, surrounded by a self-deposited basement membrane, and exposed to stiffness cues via an underlying collagen I-coated PA gel. This setup models epithelial cells breaching the basement membrane in response to external rigidity. The authors should make this organization explicit, clarifying that (1) acini possess an intact basement membrane, (2) PA gel stiffness represents tissue rigidity outside the basement membrane, and (3) collagen I functionalization enables mechanical coupling across the interface. This explanation would justify the model design and the use of collagen I-coated PA gels.

Response: We appreciate the reviewer’s clear summary and added this description in the Results section (Page 4): “In a search for novel molecular regulators linking ECM mechanical cues to EMT and invasion, we utilized a hydrogel Matrigel overlay 3D mammary acini culture system as the following: 1) the base layer is the polyacrylamide(PA) hydrogel with calibrated elastic moduli ranging from 150-320Pa present in normal human breast tissues to 1100-5700Pa observed in some stiff breast tumors; 2) the PA hydrogel is crosslinked with collagen I to enable mechanical coupling across the interface; 3) mammary acini embedded in Matrigel, which provides basement membrane-like matrices, are seeded on top of the hydrogel to be exposed to stiffness cues coming from the underlying collagen I-coated PA gel. The 3D model employed here recapitulates the biochemical and mechanical ECM microenvironment of breast carcinoma in situ, enabling mechanistic dissection of rigidity-dependent signaling events that initiate EMT. At low ECM rigidities, human basal-like non-tumorigenic MCF10A cells form mammary acini surrounded by basement membrane, while they undergo EMT and invade at high rigidities (**Fig. 1a, f and Extended Data Fig. 1b**).”